# Synthesis, Characterization, and Non-Covalent Interactions of Palladium(II)-Amino Acid Complexes

**DOI:** 10.3390/molecules26144331

**Published:** 2021-07-17

**Authors:** David B. Hobart, Michael A. G. Berg, Hannah M. Rogers, Joseph S. Merola

**Affiliations:** 1Department of Chemistry, Virginia Polytechnic Institute & State University, Blacksburg, VA 24061, USA; david.hobart@airliquide.com (D.B.H.J.); bergm@vt.edu (M.A.G.B.); hannahm@vt.edu (H.M.R.); 2Air Liquide Advanced Materials, 197 Meister Ave., Branchburg, NJ 08876, USA

**Keywords:** palladium, chelate, amino acid, hydrogen bonding, non-covalent interaction, X-ray crystallography

## Abstract

The reaction of palladium(II) acetate with acyclic amino acids in acetone/water yields square planar bis-chelated palladium amino acid complexes that exhibit interesting non-covalent interactions. In all cases, complexes were examined by multiple spectroscopic techniques, especially HRMS (high resolution mass spectrometry), IR (infrared spectroscopy), and ^1^H NMR (nuclear magnetic resonance) spectroscopy. In some cases, suitable crystals for single crystal X-ray diffraction were able to be grown and the molecular structure was obtained. The molecular geometries of the products are discussed. Except for the alanine complex, all complexes incorporate water molecules into the extended lattice and exhibit N-H···O and/or O···(HOH)···O hydrogen bonding interactions. The non-covalent interactions are discussed in terms of the extended lattice structures exhibited by the structures.

## 1. Introduction

Prior work from our group has dealt with the simplest α amino acids, the glycines [1], and the cyclic amino acids, the prolines [2], respectively. In this paper, we will explore the synthesis, characterization, and non-covalent interactions of the palladium(II) complexes of the remaining naturally occurring amino acids. Containing aliphatic R-groups, these amino acids are perhaps less unique, but no less important than their previously discussed counterparts. In this discussion of these complexes, they are grouped according to the chemical nature of the R-group of the amino acid ligand into the following five categories: aliphatic hydrophobic, aromatic hydrophobic, polar neutral, charged acidic, and charged basic.

Most of the palladium(II) complexes of the remaining naturally occurring alpha amino acids have been described to some extent in the literature dating back to the 1960s and 1970s. Many of these syntheses were only studied by a single technique and their characterization data are incomplete. A goal of our research from the outset has been to completely characterize these complexes to a degree not previously seen or reported in the literature. To this end, we have obtained ^1^H NMR data, high-resolution time-of-flight mass spectrometric data, elemental analyses, and ^13^C NMR and X-ray single crystal structures where possible. Of particular interest to us is their ability to form fascinating extended lattice structures primarily due to intermolecular hydrogen bonding, sometimes with the inclusion of water in the H-bonding motif. We recently reported on some incredible motifs in palladium complexes of beta-amino acids [3].

Among the aliphatic hydrophobic amino acids, bis-(alanino)palladium(II) was first reported by Sharma [4] in 1964 and characterized solely by infrared spectroscopy. Farooq prepared the complex in 1973 [5], characterizing the product only by potentiometric titration. In 1977 Chernova [6] also reported a synthesis of bis-(alanino)palladium(II) and characterized the product by elemental analysis and molar conductance measurements. Pletnev et al. reported the single crystal X-ray structure of bis-(valinato)palladium(II) showing the *cis* geometry [7]. Bis-(isoleucinato)palladium(II) was prepared by Patel [8] using DL-leucine in 1996 for studying pulse radiolysis and characterized only by elemental microanalysis. Hooper [9] synthesized versions of the bis-isoleucinato palladium complexes using DL-isoleucine and characterized them by infrared spectroscopy. The bis-(leucinato) and bis-(valinato) complexes were prepared by Farooq [5]. Once again, potentiometric titration was the only method of characterization reported. Jarzab [10] reported a room-temperature crystal structure of the bis-(valinato) complex in 1973, however, no hydrogen atom positions were included in the refinement, nor was any other characterization data presented in the paper.

The aromatic hydrophobic amino acid-containing palladium(II) complexes previously reported in the literature include the bis-(phenylalaninato) and bis-(tyrosinato). The crystal structure of bis-(tyrosinato)palladium(II) was reported by Jarzab [11] with no other characterization data. As before, the structure was obtained at room temperature and with no hydrogen atom positions reported. Chernova [6], using the same methods mentioned above with the alanine complex, reported the bis-(phenylalaninato) complex in 1976. You et al. also reported the structure of phenylalanine palladium(II) compounds [12].

All the complexes formed from polar neutral amino acid ligands were originally reported in the 1970s, however, the trend of sparse characterization data continues. The palladium(II) bis-chelates of asparagine, methionine, and serine were reported by Farooq [5], who again used a potentiometric titration of ligand concentration to characterize the complexes. Vagg [13] followed up with a room temperature crystal structure of the bis-(serinato) complex in 1979. Hydrogen atom positions were not reported, nor were any other data were given. Kollmann [14] reported on the synthesis of gold(III), palladium(II), and platinum(II) complexes of threonine; only melting point data were provided. Krylova [15] added to the data available for bis-(threoninato)palladium(II) with ^1^H and ^13^C NMR data for both the *cis* and *trans* isomers, as well as room temperature crystal structures. Bis-(glutaminato)palladium(II) and bis-(cysteinato)palladium(II) were both reported in 1979 by Graham [16] and Pneumatikakis [17], respectively. Graham relied solely on elemental analysis for the characterization of bis-(glutaminato)palladium(II) whereas Pneumatikakis relied on infrared spectra, elemental analysis, and solution conductivity measurements to characterize the bis-(cysteinato) complex. There is one other possible member of this group of complexes, the bis-chelate of cystine. Cystine is a derivative of cysteine, formed by the oxidation of cysteine to generate a disulfide bond (Figure 1). Complexes with this amino acid have not been reported in the literature but may exhibit some interesting reactivity given its ability to crosslink multiple metal centers into clusters or nanoparticles.

Glutamic acid and aspartic acid comprise the group of amino acids with charged acidic R-groups. The bis-chelates of these amino acids were both reported by Spacu in 1962 [18] and 1966 [19] respectively. Bis-(glutamic acid)palladium(II) was characterized by elemental analysis and based on titration stoichiometry against ethylenediamine to form the [Pd(en)_2_]^2+^ cation and against thiourea to form the [Pd(thio)_4_]^2+^ cation. Seifert reported the crystal structure confirming the normal chelation of the alpha O, N groups [20]. Bis-(aspartic acid)palladium(II) was characterized in an identical manner.

The last of the groupings of amino acids, those with charged basic R-groups, is comprised of arginine, histidine, and lysine. Of these, only bis-(histidinato)palladium(II) has been reported in the literature by Chernova [21,22]. Elemental analysis and the carboxylate symmetric and asymmetric infrared stretching vibrations are reported to confirm the product which was studied by potentiometric titration.

## 2. Results and Discussion

### 2.1. Amino Acids with Aliphatic Hydrophobic R-Groups

The first class of amino acid ligands investigated were those that possess a hydrophobic, aliphatic R-group. Alanine, valine, isoleucine, *tert*-leucine, and leucine fall into this category. Containing only aliphatic R-groups, this grouping of amino acid ligands is solely capable of *N*,*O*-chelation (Figure 2). ^1^H NMR data are consistent with a coordinated amino acid ligand, showing varying degrees of resonance shifting as compared to the free ligand. These complexes are extremely sparingly soluble in common solvents; as a result, ^13^C NMR spectra were unobtainable. In every case, high-resolution time-of-flight mass spectrometry verified the formation of the complex.

Crystals suitable for X-ray diffraction were obtained for the alanine, valine, and isoleucine complexes. The alanine complex adopts a *trans* geometry (Figure 3) much like the glycine complex when prepared from palladium(II) acetate. This suggests that the nature of the R-group exerts some influence on the coordination geometry of the complex. If the R-group is a very weakly electron-donating species the complex forms the *trans* isomer. This is born out experimentally from the glycine and alanine complexes where the R-group is a hydrogen atom or methyl group, respectively. Once the R-group takes on more of an electron-donating nature the complex tends to form the *cis* isomer upon coordination. Valine, with an isopropyl R-group, has the next highest electron donating R-group and forms the *cis* isomer. All the complexes that follow valine in order of increasing electron donating power of their R-group, for which we have X-ray crystal structures, adopt the *cis* geometry as well.

*Trans*-bis-(alaninato)palladium(II) (**1**) crystallizes in the *P*1 space group with no water molecules in the lattice. Pd-O bonds are 2.036(15) and 1.962(14) Å and Pd-N bond lengths are 2.02(2) and 2.04(2) Å. N-Pd-O bond angles within chelate rings are 80.6(8)° and 83.1(8)°, and the N-Pd-O bond angles between chelate rings are 99.0(7)° and 97.3(7)°. These values compare favorably with other palladium(II) amino acid chelates previously reported [1,6,10,11,13,23,24]. Intermolecular hydrogen bonding is observed between an amine proton and carbonyl oxygen on adjacent molecules in the lattice (Figure 4).

*Cis*-bis-(l-valinato)palladium(II) (**2**) crystallizes in the *P*2_1_2_1_2_1_ space group with one water molecule in the lattice (Figure 5). The Pd-N bonds are 2.0054(15) and 2.0110(16) Å, and the Pd-O bonds are 2.0054(15) and 2.0170(13) Å. The O-Pd-O bond angle is 96.03(6)°, and the N-Pd-N bond angle is 96.90(6)°. These values are comparable to values reported for other square planar palladium(II) chelates [1,6,10,11,13,23,24]. Intermolecular hydrogen bonding is observed in the crystal lattice (Figure 6). An amine proton on each of the nitrogen atoms of one complex molecule is hydrogen bonded to a carbonyl oxygen and a chelated carboxylate oxygen on the adjacent molecule. Both amine hydrogen atoms lie on the same side of the chelate plane. Additionally, the water molecule in the lattice is hydrogen bonded between two carbonyl oxygen atoms on adjacent complex molecules and between the other two amine protons on yet a third complex molecule in the lattice (Figure 6).

The last complex from this group for which a crystal structure was able to be determined was *cis*-bis-(isoleucinato)palladium(II) (**4**) (Figure 7). Crystallizing in the *P*2_1_2_1_2_1_ space group, it has Pd-N bond lengths of 2.008(3) and 2.008(3) Å and Pd-O bond lengths of 2.014(2) and 1.997(3) Å. As before, these values compare favorably with the literature [1,6,10,11,13,23,24]. There is one water molecule in the lattice. The R-groups are oriented such that one lies on either side of the chelate plane.

The extended lattice observed for this complex is similar to that seen for the valine complex and involves hydrogen bonding between amine protons on both nitrogens of one complex molecule to both oxygen atoms in one chelate ring on an adjacent molecule in the lattice (Figure 8). The water molecule in the lattice is hydrogen bonded between the two carbonyl oxygen atoms on adjacent complex molecules and in this case, is not hydrogen bonded to the remaining amine hydrogen atoms.

Bis-(*tert*-leucinato)palladium(II) (**5**) and bis-(leucinato)palladium(II) (**6**) were also prepared, and although crystals were not able to be grown, the HRMS and NMR data are consistent with the formation of the bis-chelates.

### 2.2. Amino Acids with Aromatic Hydrophobic R-Groups

Tryptophan, phenylalanine, and tyrosine comprise the group of amino acids with aromatic R-groups; their proposed palladium bis-chelates are shown in Figure 9.

Phenylalanine and tyrosine can only chelate in an *N*,*O* manner, as they lack another coordinating moiety on their R-groups. An argument could be made for *O*,*O* chelation with tyrosine, but that is extremely unlikely. *O*,*O* chelation would result in a 9-membered chelate ring which would be considerably less stable than the 5-membered chelate formed via *N*,*O* chelation. Sabat confirmed the *N*,*O* chelation and an X-ray crystal structure indicates the *trans* geometry of the bis-chelate [24]. Additionally, the incorporation of a planar phenyl ring into the chelate ring is sterically very unfavorable. Tryptophan, on the other hand, does present an intriguing option for *N*,*N* chelation resulting in a 7-membered chelate ring. While not as stable as the 5-membered *N*,*O* chelate, it is nonetheless possible as evidenced by Nakayama’s preparation of the ornithine complex [25]. In this case, we can turn to an interpretation of the infrared spectra of the chelated and free ligand to determine the coordination mode (Figure 10). The FTIR spectrum of the free ligand shows an asymmetric CO_2_ stretch at 1573 cm^−1^ and a symmetric CO_2_ stretch at 1406 cm^−1^. This is expected when the carboxylate is in the zwitterionic state. If the carboxylate is coordinated to the palladium center, the asymmetric and symmetric stretching modes coalesce into a single carbonyl stretching vibration shifted to higher frequency. This is exactly what is observed with bis-(tryptophanato)palladium(II) (**7**). The FTIR spectrum of the complex shows that the asymmetric and symmetric stretching vibrations are absent and a new carbonyl stretch is present at 1647 cm^−1^. In addition, the indole N-H stretch at 3407 cm^−1^ is unchanged between the free ligand and complex. Were tryptophan to coordinate through the indole nitrogen atom, a shift in the indole N-H stretch would be observed. On the basis of this evidence we can say with some certainty that bis-(tryptophanato)palladium(II) (**7**) is a 5-membered *N*,*O* chelate, however the *cis-trans* geometry of the ligands about the metal center remains undetermined. NMR and HRMS data for these compounds are as expected and unremarkable. They are summarized for each compound in the Appendix A.

### 2.3. Amino Acids with Polar Neutral R-Groups

Neutral, polar side groups characterize the next grouping of Pd-AA complexes (Figure 11). These amino acids include asparagine, glutamine, cysteine, methionine, serine, and threonine.

Bis-(asparaginato)palladium(II) (**10**) and bis-(glutaminato)palladium(II) (**11**) both contain an amide group on the R-group terminus and differ only by one methylene unit in the side chain. HRMS confirms the formation of the bis-chelate in both instances, and the NMR data for each complex are as expected. As neither of these techniques speaks to the chelation geometry, we must again turn to the FTIR data to determine whether chelation is *N*,*N* or *N*,*O* in nature. Free asparagine shows a symmetric CO_2_ stretch at 1394 cm^−1^ and an asymmetric stretch at 1495 cm^−1^. Both of these vibrations are missing in the spectrum of the complex with a new CO stretch appearing at 1688 cm^−1^. Furthermore, the amide carbonyl stretching vibrations at 1601 and 1642 cm^−1^ remain unchanged in the spectrum of the complex. This is indicative of *N*,*O* chelation for bis-(asparaginato)palladium(II) (**10**). This trend is seen as well for bis-(glutaminato)palladium(II) (**11**) with symmetric and asymmetric CO_2_ stretching vibrations at 1445 and 1572 cm^−1^, respectively, disappearing from the spectrum of the chelate. A new CO stretch at 1636 cm^−1^ is observed in the spectrum of the complex. The amide CO stretching vibrations at 1327 and 1408 cm^−1^ remain unchanged in both the free ligand and the complex. Once again this is indicative of *N*,*O* chelation for bis-(glutaminato)palladium(II) (**11**).

Serine and threonine both contain hydroxyl group functionality in their R-groups that lead to extended hydrogen bonded networks in the solid phase. *Cis*-bis-(serinato)palladium(II) (**15**) (Figure 12) crystallizes in the *P*2_1_2_1_2_1_ space group. Pd-N bond lengths are 2.0173(19) and 2.0114(18) Å with a N-Pd-N bond angle of 100.28°. Pd-O bond lengths are 2.0175(16) and 1.9981(16) Å with an O-Pd-O bond angle of 94.21(6)° and an N-Pd-N angle of 100.23(7)°. These values are comparable to other square planar palladium(II) chelates reported in the literature [1,6,10,11,13,23,24].

The extended lattice of (**15**) is extremely well ordered, with ten hydrogen bonding interactions present for each molecule. Each molecule in the lattice is hydrogen bonded to six other molecules within the lattice (Figure 13).

*Cis*-bis-(threoninato)palladium(II) (**16**) (Figure 14) differs from the serine complex by the replacement of one side-chain methylene hydrogen atom with a methyl group. This complex also crystallizes in the *P*2_1_2_1_2_1_ space group with 2 complex molecules and 3 water molecules in the unit cell. Pd-N bond lengths are 2.015(4) and 2.009(4) Å for one independent molecule and 2.010(4) and 2.013(4) Å for the second molecule. Pd-O bond lengths are 2.012(3) and 2.029(3) Å for one molecule and 2.009(3) and 2.023(3) Å for the second independent molecule. Bond angles for the two independent molecules are 98.39(16)° and 97.66(15)° for the N-Pd-N angles and 95.88(13)° and 96.85(13)° for the O-Pd-O angles. Once again, these values are in line with other square planar palladium(II) chelates reported in the literature [1,6,10,11,13,23,24]. As seen with the serine complex, there is a very well-ordered hydrogen bonding network present within the extended lattice for this complex (Figure 15). Each complex molecule in the lattice is involved in one intermolecular hydrogen bond from the side chain hydroxyl hydrogen atom to the coordinated carboxylate oxygen atom of another complex molecule. There are an additional 3 hydrogen bonds to water molecules in the lattice that bridge to three other distinct complex molecules in the lattice.

Of the six amino acid ligands in this group, methionine and cysteine contain a sulfur atom in their molecular structure. A cysteine derivative, cystine, contains a disulfide group and will be discussed here as well. Sulfur-containing ligands are an interesting choice for palladium(II) chemistry in that palladium has a noted affinity for sulfur over nitrogen and oxygen groups. Incorporation of the sulfur moiety into an amino acid ligand structure provides one method of forming multi-dentate network structures. The first sulfur-containing amino acid complex of palladium(II) that was attempted to be synthesized was bis(methioninato)palladium(II) (**14**). It was hoped that the thioether side chain would be less likely to coordinate than the thiol group present in cysteine. High–resolution time-of-flight mass spectrometry performed on a sample taken directly from the reaction mixture clearly showed the formation of a bis-methionine compound (Figure 16). HRMS is extremely useful in the characterization of any compound, but especially for palladium complexes [26]. In addition to obtaining the exact mass of fragments that allow for a clear determination of stoichiometry, the fragmentation patterns uniquely identify palladium compounds. The isotope distribution can definitively confirm or exclude the presence of other elements such as sulfur or chlorine. The manifold for compound (**14**) as shown in Figure 16 is unambiguous—while HRMS/esi cannot confirm the nature of the chelation (*N*,*O* vs. *N*,*S*) it does confirm that each ligand is bidentate to the palladium. The HRMS/esi+ spectra for the compounds described in this paper are available in the Appendix A.

Frustratingly, all subsequent attempts to isolate the bis-chelated product failed. NMR spectra were unable to be obtained, as were single crystals for X-ray diffraction. FTIR spectra were inconclusive and appeared to show a mixture of methionine ligand and palladium(II) acetate.

Battaglia et al. described the crystal structure of S-methyl cysteine palladium dichloride with *N*,*S* coordination, and a free carboxylic acid side chain [27]. In our hands, the reaction of cysteine with palladium(II) acetate to yield bis-(cysteinato)palladium(II) (**12**) also failed to yield an isolatable product, however, the HRMS data from the reaction mixture show a series of products whose masses correspond exactly to discreet ratios of ligand and metal. In these proposed structures (Figure 17) each palladium center is *N*,*O* chelated by the ligand, with the thiol group and the non-coordinated carboxylate carbonyl oxygen coordinated to another palladium center.

In this case di-, tri-, and tetra-nuclear palladium(II) complexes were detected. The synthesis of bis-(cystinato)palladium(II) (**13**) was also attempted from cystine and palladium(II) acetate, and once again failed to yield an isolatable product. Palladium cluster aggregation is a noted phenomenon with thioethers [28] and disulfides [29] and likely accounts for the lack of product formation in the cases of methionine and cystine.

### 2.4. Amino Acids with Charged Acidic R-Groups

Aspartic acid and glutamic acid comprise the ligand set composed of acidic R-groups (Figure 18). Here again, we see the possibility of multiple coordination modes. *N*,*O* chelation forms the more stable 5-membered chelate ring, however *O*,*O* chelation is possible and would yield 6-membered and 7-membered chelate rings, respectively. Studies on donor atom preferences for palladium have clearly established that palladium prefers ligands containing sulfur over nitrogen, and nitrogen over oxygen (S > N > O). On this reasoning alone we would expect the palladium complexes of aspartic and glutamic acids to adopt the 5-membered *N*,*O* chelate ring. Fortuitously, the crystal structure of *cis*-bis-((η^2^-*N*,*O*)-aspartate)palladium(II) (**17**) was obtained (Figure 19) and clearly shows that the chelation mode is *N*,*O*. It is not unreasonable to expect that glutamic acid would behave in the same manner.

*Cis*-bis-((η^2^-*N*,*O*)-aspartate)palladium(II) (**17**) crystallizes in the *P*2_1_ space group and exhibits an extensive hydrogen bonding network (Figure 20). Pd-O bond distances are 2.037(3)Å and 2.014(3)Å while Pd-N bond distances are 2.009(3)Å and 2.008(3)Å with an O-Pd-O angle of 9.30(11)° and an N-Pd-N angle of 98.07(12)°. Two amine protons that lie on one side of the chelate plane are each hydrogen bonded to separate coordinated carboxyl oxygen atoms on an adjacent molecule. The two amine protons on the other side of the chelate plane are hydrogen bonded to the R-group aspartate carboxyl group of another adjacent molecule. Additionally, there are three hydrogen bonded water molecules in the lattice that bridge the coordinated carbonyl oxygen atom on one molecule to the coordinated carbonyl oxygen atom on another molecule.

### 2.5. Amino Acids with Charged Basic R-Groups

Basic R-groups are present in the palladium(II) complexes of histidine, arginine, and lysine (Figure 21).

*N*,*N* chelation modes are possible for all of these complexes. For bis-(lysinato)palladium(II) (**21**) *N*,*N* chelation would result in an unlikely 8-membered chelate ring. This leaves *N*,*O* chelation as the only possible and most favored coordination mode. Due to the complexity of the fingerprint region (1500–500 cm^−1^) of the FTIR spectrum for this complex, a definite assessment based on the carboxylate stretching vibrations cannot be made and attempts to grow crystals for X-ray diffraction were unsuccessful. The ^1^H and ^13^C NMR spectra are consistent with a coordinated lysine ligand, and HRMS confirms the formation of bis-(lysinato)palladium(II) (**21**).

Bis-(argininato)palladium(II) (**20**) contains a guanidine group on the terminus of the amino acid R-group that could form either a 7-membered or 9-membered chelate ring. The 9-membered ring is not likely to form, however as discussed above for tryptophan the possibility of a 7-membered ring cannot be dismissed out of hand. Fortunately, the FTIR spectrum of the guanidine side chain of arginine is well-characterized and there are several characteristic C-N stretches that indicate the presence of an un-coordinated guanidine group. Infrared absorptions at 920, 1180, 1614, and 1670 cm^−1^ have been assigned to various C-N stretching modes of the free guanidine group. These absorptions are present in both the free ligand and the complex (Figure 22) and support the premise that the arginine side chain is not involved in coordination with the metal. Proton NMR data are consistent with a coordinated arginine ligand and HRMS confirms the formation of the bis-chelate.

Histidine (Figure 23) provides an opportunity for *N*,*N* chelation with the formation of a 6-membered ring. The R-group terminus of the histidine molecule is an imidazole group.

Coordination through N-1 is unlikely as the lone pair of electrons on this nitrogen atom are involved in the aromatic π system of the imidazole ring. Coordination through N-3 is possible and would result in the formation of a 6-membered *N*,*N* chelate, a chelation mode we observed in a Cp*Ir(his)Cl complex [30]. Turning again to the FTIR spectrum of the free ligand versus the complex, the symmetric CO_2_ stretch in the free ligand at 1449 cm^−1^ disappears and is not observed in the complex. The asymmetric CO_2_ stretch at 1628 cm^−1^ in the free ligand is shifted to 1588 cm^−1^ in the complex. This is compelling evidence of the lack of free carboxylate functionality in the complex and strongly indicates the formation of the *N*,*O* chelate. HRMS confirms the formation of bis-(histidinato)palladium(II) (**19**). Interestingly, the NMR data indicate the presence of two distinct complexes in D_2_O solution. Each of the expected six carbon resonances in the ^13^C NMR spectrum appears as a pair of resonances. Similarly, the ^1^H NMR spectrum shows two sets of resonances for the methine and aromatic protons that each integrate to a 1:1:1 ratio. The methylene protons show significant overlap and cannot be resolved, but they integrate into a total of 4 protons. It is very likely that we are seeing an aquo-complex formation in solution as we observed for the glycine complexes discussed in a previous paper [1].

## 3. Solid State Intermolecular Interactions

To a large degree, our interest in amino acid complexes of metals stems from their ability to make intricate intermolecular networks due to hydrogen-bonding. The role that H-bonding plays in proteins is enormous and H-bonding in metal complexes of amino acids can also play a huge role in crystal formation and crystal engineering, especially when involving waters of hydration [31,32,33]. Infantes et al. examined the crystal literature and described various motifs they found [34,35]. The number of possible hydrogen bonding motifs is quite large and Infantes et al. have proposed a system to describe these complex motifs. Developing a deeper understanding of H-bonding in organic and organometallic systems is a complex and ongoing field of study [31,32,36,37,38].

All the compounds described above characterized by single crystal X-ray diffraction display hydrogen bonding that sets up a definite network throughout the crystal lattice. It has been our experience that this network often stabilizes the crystals and prevents waters of crystallization (when included) from being lost even when the crystals are subject to heat and vacuum.

The lattice of complex (**1**) incorporates no additional waters, but intermolecular H-bonding between the individual molecules sets up a strong network of N-H···O hydrogen bonding that results in offset layers of complexes.

Complex (**1**) has a *trans* geometry of the alanine ligands and the methyl group is not very sterically demanding and so this network is purely molecule-to-molecule. Figure 24 shows the offset layers of hydrogen bonding that make up the more-or-less two-dimensional network. A different view is seen in Figure 25 that more clearly shows that the hydrogen-bonding occurs within two-dimensional layers while those layers, in turn, are weakly bonded to neighboring layers with purely van der Waals interactions. So, for compound 1, that combination of hydrogen-bonding and van der Waals forces comprise the non-covalent interactions for this lattice.

On the other hand, the valine complex (**2**), is of cis geometry with a bulky isopropyl side chain. For these reasons, direct molecule-to-molecule interactions are hindered and so a water of hydration is retained in the lattice and the H-bonding network creates a 3-dimensional pattern rather than the 2-D layers seen in the alanine compound (Figure 26). This shows a rather complex network but referring to Figure 6, one can see that the “core” of this structure has the water of hydration playing a pivotal H-bonding role. The water assumes a hydrogen-donor role to the non-coordinate carboxylate oxygen atom of two molecules of complex and a hydrogen-acceptor role to an N-H bond of a third molecule of the complex. Within this complex triad, additional H-bonding occurs between the two molecules that act as H-bond acceptor to the water. This core triad then continues H-bonding interactions to build up the entire crystal. These non-covalent interactions make for a strongly bonded 3-dimensional lattice.

For complex (**4**), the isobutyl group has similar steric demands to the isopropyl group and the H-bonding network in its crystal lattice is quite different from complex (**2**). Again, a lattice water molecule serves as an H-bond donor to a non-bonded carboxylate oxygen, but there is no analogous H-bond acceptor role from an N-H bond. Instead, weaker, but still recognizable C-H···O bonds are found. C-H···O bonds have been recognized since 1937 and crystallographic evidence was discussed at some length by Sutor in 1962 [39], although acceptance of such a weak bond was not initially forthcoming but now they are recognized, especially in many biomolecules [40]. The C···O distance in complex 4 is 3.207(11) Å, just at the limit of what is accepted to be a C-H···O hydrogen bond [40].

For the complexes with additional heteroatoms on side chains, those side-chain atoms can provide additional sites for non-bonded interatomic interactions. The serine complex, (**15**), does not incorporate any additional waters but the -OH group on the side chain participates in the bonding network, also giving a complicated 3-D arrangement of molecules. With six different sites (O atoms) for hydrogen bond acceptance and six different sites for hydrogen bond donation (4 N-H and one O-H), the H-bonding interactions in the crystal lattice is complex indeed. In addition to Figure 13 above, see Figure 27 with another view of the numerous non-covalent H-bonding interactions throughout the lattice.

The addition of more steric bulk on the side change of the threonine complex, (**16**), does hinder some of the interactions in the serine compound and so the lattice incorporates a molecule of water. The role of water in many of these compounds is similar, but the types and number of non-bonded interactions are quite different overall. For the threonine complex (**16**), there are potentially 4 interactions possible (2 donating, 2 accepting) but only 3 are found (Figure 28). In this case, there are 2 H-bond donor interactions: one to the non-coordinated carboxylate oxygen atom of one molecule and to the sidechain OH of a second. The water only acts as an acceptor for the OH bond of a third complex molecule. While there is a somewhat short C-H···O interaction, that interaction at 3.529(6) Å is well above the recognized maximum distance of 3.2 Å for such a weak bond [40].

Finally, complex (**17**) with the aspartic acid ligand again has the inclusion of a water of hydration. Combined with the non-coordinated carboxylic acid groups, a layered motif is established with alternating layers of the Pd centers of the molecules, and layers of water and the dangling carboxylic acids, (Figure 29).

All 6 of the structures discussed in this manuscript show an abundance of non-covalent intermolecular interactions including O-H and N-H hydrogen bond donation, and O hydrogen bond accepting character to a number of types of O atoms: both bonded and non-bonded carboxylate oxygen atoms, sidechain O-H, and -COOH oxygen atoms and lattice water oxygen atoms. The complexity of these interactions is similar to that found in proteins and it is little wonder that amino acid complexes of metals display a range of biological activity [41].

## 4. Materials and Methods

All materials were purchased from commercial sources and used without further purification. The ^1^H NMR spectra were collected on a Varian MR-400 NMR spectrometer (Agilent, Santa Clara, CA 95051, USA) or a Bruker Avance III 600 MHz NMR spectrometer (Billerica, MA 01821, USA). ^13^C NMR spectra were collected on a Bruker Avance III 600 MHz NMR spectrometer. High Resolution Mass Spectra (HRMS/ESI+) were collected on an Agilent 6220 Accurate Mass TOF MS with an ESI source (Agilent, Santa Clara, CA 95051, USA). Solid-state Fourier Transform Infrared (FTIR) spectra were collected on a Midac M2000 FTIR spectrometer equipped with a DuraScope diamond ATR accessory (Midac Corporatyion, Westfield, MA 01085, USA). X-ray crystallographic data were collected at 100 K on an Oxford Diffraction Gemini diffractometer with an EOS CCD detector and Mo Kα radiation (Rigaku Americas Corporation, Woodlands, TX 77381, USA).

### 4.1. General Procedure

The following procedure was followed for all complex syntheses reported in this paper:

A four-dram vial was charged with palladium(II) acetate and 3.0 mL of 50/50 (*v*/*v*) acetone/water. The mixture was stirred until completely dissolved. To this was added the appropriate amino acid and left to stir overnight. All reaction solutions turned from a dark red orange to a clear yellow with a pale yellow-white precipitate on overnight stirring. An odor of acetic acid was noted when the vial was opened. The supernatant was transferred via pipette to a clean vial and allowed to evaporate to give clear yellow needles sometimes suitable for X-ray crystallography. The pale-yellow precipitate was washed with cold water and dried under vacuum. The original reaction precipitate and the material obtained from evaporation were combined and weighed to determine a final yield.

#### 4.1.1. Synthesis of Bis-(alaninato)palladium(II) (**1**)

Following the standard procedure, 0.0532 g palladium(II) acetate (2.37 × 10^−4^ mol) and 0.0464 g l-alanine (5.21 × 10^−4^ mol, 2.2 equivalents) were combined. The combined yield of crystals and precipitate was 63.3 mg of product (0.2240 mmol, 95% yield). Pd(C_3_H_6_NO_2_)_2_ was identified based on the following data: ^1^H NMR (400 MHz, Deuterium Oxide) δ 3.63 (q, *J* = 7.2 Hz, 1H), 1.33 (d, *J* = 7.3 Hz, 3H). HRMS/ESI+ (*m*/*z*): [M + H]^+^ calcd for Pd(C_3_H_6_NO_2_)_2_, 282.9905; found, 282.9916.

#### 4.1.2. Synthesis of Bis-(l-valinato)palladium(II) (**2**)

Following the standard procedure, 0.0256 g palladium(II) acetate (1.14 × 10^−4^ mol) and 0.0294 g l-valine (2.51 × 10^−4^ mol, 2.2 equivalents)were combined. The combined yield of crystals and precipitate was 36.6 mg of product (0.1081 mmol, 95% yield). Pd(C_5_H_10_NO_2_)_2_ was identified based on the following data: ^1^H NMR (400 MHz, Deuterium Oxide) δ 3.47 (d, *J* = 4.4 Hz, 1H), 2.14 (pd, *J* = 7.0, 4.4 Hz, 1H), 0.88 (dd, *J* = 20.8, 7.0 Hz, 6H). HRMS/ESI+ (*m*/*z*): [M + H]^+^ calcd for Pd(C_5_H_10_NO_2_)_2_, 339.0531; found, 339.0549.

#### 4.1.3. Synthesis of Bis-(d-valinato)palladium(II) (**3**)

Following the standard procedure, 0.0188 g palladium(II) acetate (8.37 × 10^−5^ mol) 0.0216 g d-valine (1.84 × 10^−4^ mol, 2.2 equivalents) were combined. 27.1 mg of product (0.0800 mmol, 96% yield) was obtained. Pd(C_5_H_10_NO_2_)_2_ was identified based on the following data: ^1^H NMR (400 MHz, Deuterium Oxide) δ 3.47 (d, *J* = 4.4 Hz, 1H), 2.14 (pd, *J* = 7.0, 4.4 Hz, 1H), 0.88 (dd, *J* = 20.8, 7.0 Hz, 6H). HRMS/ESI+ (*m*/*z*): [M + H]^+^ calcd for Pd(C_5_H_10_NO_2_)_2_, 339.0531; found, 339.0557. Anal. Calcd. for Pd(C_5_H_10_NO_2_)_2_·H_2_O: C, 33.67%; H, 6.22%; N, 7.85%. Found: C, 33.83%; H, 6.29%; N, 8.01%.

#### 4.1.4. Synthesis of Bis-(isoleucinato)palladium(II) (**4**)

Following the standard procedure, 0.1638 g palladium(II) acetate (7.30 × 10^−4^ mol) and 0.2105 g isoleucine (1.61 × 10^−3^ mol, 2.2 equivalents) were combined. The yield of product was 254.9 mg (0.6950 mmol, 95% yield). Pd(C_6_H_12_NO_2_)_2_ was identified based on the following data: ^1^H NMR (500 MHz, DMSO-*d*_6_) δ 5.15 (dd, *J* = 10.5, 7.6 Hz, 1H), 4.72 (t, *J* = 8.9 Hz, 1H), 4.37 (dd, *J* = 10.6, 5.5 Hz, 1H), 3.84 (dd, *J* = 10.7, 5.2 Hz, 1H), 3.16–3.02 (m, 1H), 1.76 (q, *J* = 10.4, 9.0 Hz, 1H), 1.57–1.31 (m, 2H), 1.04 (d, *J* = 6.9 Hz, 3H), 0.86 (dt, *J* = 22.8, 7.4 Hz, 3H). HRMS/ESI+ (*m*/*z*): [M + H]^+^ calcd for Pd(C_6_H_12_NO_2_)_2_, 367.0844; found, 367.0871.

#### 4.1.5. Synthesis of Bis-(tert-leucinato)palladium(II) (**5**)

Following the standard procedure, 0.0325 g palladium(II) acetate (1.45 × 10^−4^ mol) and *tert*-leucine (3.18 × 10^−4^ mol, 2.2 equivalents) were combined. The yield of product was 50.6 mg (0.1378 mmol, 95% yield). Pd(C_6_H_12_NO_2_)_2_ was identified based on the following data: ^1^H NMR (400 MHz, Deuterium Oxide) δ 3.16 (s, 1H), 0.94 (s, 9H). HRMS/ESI+ (*m*/*z*): [M + H]^+^ calcd for Pd(C_6_H_12_NO_2_)_2_, 367.0844; found, 367.0877. Anal. Calcd. for Pd(C_6_H_12_NO_2_)_2_: C, 39.30%; H, 6.60%; N, 7.64%. Found: C, 39.47%; H, 6.59%; N, 7.70%.

#### 4.1.6. Synthesis of Bis-(leucinato)palladium(II) (**6**)

Following the standard procedure, 0.0517 g palladium(II) acetate (2.30 × 10^−4^ mol) and 0.0665 g leucine (5.07 × 10^−4^ mol, 2.2 equivalents) were combined. The yield of product was 82.1 product (0.2239 mmol, 97% yield). Pd(C_6_H_12_NO_2_)_2_ was identified based on the following data: ^1^H NMR (400 MHz, Deuterium Oxide) δ 3.63–3.53 (m, 1H), 1.65–1.55 (m, 1H), 1.60–1.44 (m, 2H), 0.90–0.72 (m, 6H). ^13^C NMR (101 MHz, D_2_O) δ 175.70, 53.57, 39.94, 24.30, 22.16, 21.02. HRMS/ESI+ (*m*/*z*): [M + Na]^+^ calcd for Pd(C_6_H_12_NO_2_)_2_, 389.0663; found, 389.0692.

#### 4.1.7. Synthesis of Bis-(tryptophanato)palladium(II) (**7**)

Following the standard procedure, 0.0799 g palladium(II) acetate (3.56 × 10^−4^ mol) and 0.1599 g tryptophan (7.83 × 10^−4^ mol, 2.2 equivalents) were combined The yield of product was 178.2 mg (0.3475 mmol, 98% yield). Pd(C_11_H_11_N_2_O_2_)_2_ was identified based on the following data: ^1^H NMR (400 MHz, DMSO-d6) δ 7.59–6.84 (m, 5H), 4.69–3.73 (m, 2H), 3.47–3.33 (m, 1H). HRMS/ESI+ (*m*/*z*): [M + H]^+^ calcd for Pd(C_11_H_11_N_2_O_2_)_2_, 513.0749; found, 513.0700. Anal. Calcd. for Pd(C_11_H_11_N_2_O_2_)_2_·H_2_O: C, 49.77%; H, 4.56%; N, 10.55%. Found: C, 49.00%; H, 4.51%; N, 10.41%.

#### 4.1.8. Synthesis of Bis-(phenylalaninato)palladium(II) (**8**)

Following the standard procedure, 0.1067 g palladium(II) acetate (4.75 × 10^−4^ mol) and 0.1727 g phenylalanine (1.05 × 10^−3^ mol, 2.2 equivalents) were combined. The product yield was 190.1 mg (0.4373 mmol, 92% yield). Pd(C_9_H_10_NO_2_)_2_ was identified based on the following data: ^1^H NMR (400 MHz, Deuterium Oxide) δ 7.34–7.14 (m, 5H), 3.89–3.82 (m, 1H), 3.15 (dd, *J* = 14.5, 5.2 Hz, 1H), 2.98 (dd, *J* = 14.5, 8.0 Hz, 1H). HRMS/ESI+ (*m*/*z*): [M + H]^+^ calcd for Pd(C_9_H_10_NO_2_)_2_, 435.0531; found, 435.0538.

#### 4.1.9. Synthesis of Bis-(tyrosinato)palladium(II) (**9**)

Following the standard procedure, 0.0511 g palladium(II) acetate (2.28 × 10^−4^ mol) and 0.0907 g tyrosine (5.01 × 10^−4^ mol, 2.2 equivalents) were combined. The product yield was 92.1 mg (0.1973 mmol, 87% yield). Pd(C_9_H_10_NO_3_)_2_ was identified based on the following data: ^1^H NMR (400 MHz, Deuterium Oxide) δ 7.11–7.04 (m, 2H), 6.83–6.73 (m, 2H), 3.82 (dd, *J* = 7.8, 5.2 Hz, 1H), 3.09 (dd, *J* = 14.7, 5.2 Hz, 1H), 2.94 (dd, *J* = 14.7, 7.8 Hz, 1H). HRMS/ESI+ (*m*/*z*): [M + H]^+^ calcd for Pd(C_9_H_10_NO_3_)_2_, 467.0429; found, 467.0448.

#### 4.1.10. Synthesis of Bis-(asparaginato)palladium(II) (**10**)

Following the standard procedure, 0.0259 g palladium(II) acetate (1.15 × 10^−4^ mol) and 0.0335 g asparagine (2.54 × 10^−4^ mol, 2.2 equivalents) were combined. The yield of product was 39.7 mg (0.1077 mmol, 93% yield). Pd(C_4_H_7_N_2_O_3_)_2_ was identified based on the following data: ^1^H NMR (400 MHz, Deuterium Oxide) δ 3.93–3.79 (m, 1H), 2.88–2.63 (m, 2H). HRMS/ESI+ (*m*/*z*): [M + H]^+^ calcd for Pd(C_4_H_7_N_2_O_3_)_2_, 369.0021; found, 369.0050. Anal. Calcd. for Pd(C_4_H_7_N_2_O_3_)_2_: C, 26.06%; H, 3.83%; N, 15.20%. Found: C, 26.20%; H, 3.72%; N, 15.28%.

#### 4.1.11. Synthesis of Bis-(glutaminato)palladium(II) (**11**)

Following the standard procedure, 0.0218 g palladium(II) acetate (9.71 × 10^−5^ mol) and 0.0312 g glutamine (2.14 × 10^−4^ mol, 2.2 equivalents) were combined. The yield of product was 34.7 mg (0.0877 mmol, 90% yield). Pd(C_5_H_9_N_2_O_3_)_2_ was identified based on the following data: ^1^H NMR (500 MHz, Deuterium Oxide) δ 3.50 (dt, *J* = 14.5, 6.5 Hz, 1H), 2.52–2.33 (m, 2H), 2.02 (ddq, *J* = 82.6, 15.0, 7.6, 6.9 Hz, 2H). HRMS/ESI+ (*m*/*z*): [M + Na]^+^ calcd for Pd(C_5_H_9_N_2_O_3_)_2_, 419.0153; found, 419.0193. Anal. Calcd. for Pd(C_5_H_9_N_2_O_3_)_2_: C, 30.28%; H, 4.57%; N, 14.12%. Found: C, 30.55%; H, 4.55%; N, 14.27%.

#### 4.1.12. Attempted Synthesis of Bis-(cysteinato)palladium(II) (**12**)

Following the standard procedure, 0.0466 g palladium(II) acetate (2.08 × 10^−4^ mol) and 0.0553 g cysteine (4.57 × 10^−4^ mol, 2.2 equivalents) were combined. HRMS/ESI+ of the supernatant shows masses of 573.8426, 798.7509, and 1024.6613 amu. No further characterization was possible.

#### 4.1.13. Attempted Synthesis of Bis-(cystinato)palladium(II) (**13**)

Following the standard procedure, 0.0349 g palladium(II) acetate (1.55 × 10^−4^ mol) and 0.0822 g cysteine (3.42 × 10^−4^ mol, 2.2 equivalents) were combined. HRMS/ESI+ of the supernatant was inconclusive for Pd(II) species. No further characterization was possible.

#### 4.1.14. Attempted Synthesis of Bis-(methioninato)palladium(II) (**14**)

Following the standard procedure, 0.0203 g palladium(II) acetate (9.04 × 10^−5^ mol) and 0.0297 g methionine (1.99 × 10^−4^ mol, 2.2 equivalents) were combined. HRMS/ESI+ (*m*/*z*): [M + H]^+^ calcd for Pd(C_5_H_10_NO_2_S)_2_, 402.9972; found, 402.9998. No further characterization was possible.

#### 4.1.15. Synthesis of Bis-(serinato)palladium(II) (**15**)

Following the standard procedure, 0.1063 g palladium(II) acetate (4.73 × 10^−4^ mol) 0.1095 g serine (1.04 × 10^−3^ mol, 2.2 equivalents) were combined. The combined yield of single crystals and precipitate was 144.9 mg of product (0.4606 mmol, 97% yield). Pd(C_3_H_6_NO_3_)_2_ was identified based on the following data: ^1^H NMR (400 MHz, Deuterium Oxide) δ 3.82–3.71 (m, 1H), 3.70–3.48 (m, 2H). HRMS/ESI+ (*m*/*z*): [M + H]^+^ calcd for Pd(C_3_H_6_NO_3_)_2_, 314.9803; found, 314.9818.

#### 4.1.16. Synthesis of Bis-(threoninato)palladium(II) (**16**)

Following the standard procedure, 0.0236 g palladium(II) acetate (1.05 × 10^−4^ mol) and and 0.0275 g threonine (2.31 × 10^−3^ mol, 2.2 equivalents) were combined. The combined yield of single crystals and precipitate was 35.0 mg of product (0.1021 mmol, 97% yield). Pd(C_4_H_8_NO_3_)_2_ was identified based on the following data: ^1^H NMR (400 MHz, Deuterium Oxide) δ 4.11–4.03 (m, 1H), 3.41 (dd, *J* = 4.9, 1.9 Hz, 1H), 1.14 (dd, *J* = 6.6, 1.9 Hz, 3H). HRMS/ESI+ (*m*/*z*): [M + H]^+^ calcd for Pd(C_4_H_8_NO_3_)_2_, 343.0116; found, 343.0152.

#### 4.1.17. Synthesis of Bis-(aspartic acid)palladium(II) (**17**)

Following the standard procedure, 0.1233 g palladium(II) acetate (5.49 × 10^−4^ mol) and 0.1608 g aspartic acid (1.21 × 10^−3^ mol, 2.2 equivalents) were combined. The combined yield of single crystals and precipitate was 192.4 mg of product (0.5191 mmol, 95% yield). Pd(C_4_H_6_NO_4_)_2_ was identified based on the following data: ^1^H NMR (400 MHz, Deuterium Oxide) δ 3.94 (dd, *J* = 7.0, 4.6 Hz, 1H), 2.93–2.80 (m, 2H). HRMS/ESI+ (*m*/*z*): [M + H]^+^ calcd for Pd(C_4_H_6_NO_4_)_2_, 370.9701; found, 370.9739.

#### 4.1.18. Synthesis of Bis-(glutamic acid)palladium(II) (**18**)

Following the standard procedure, 0.0194 g palladium(II) acetate (8.64 × 10^−5^ mol) and 0.0280 g glutamic acid (1.90 × 10^−4^ mol, 2.2 equivalents) were combined. The yield of product was 32.1 mg (0.0805 mmol, 93% yield). Pd(C_5_H_8_NO_4_)_2_ was identified based on the following data: ^1^H NMR (400 MHz, DMF-d7) δ 3.47 (s, 1H), 2.96–2.88 (m, 2H), 2.79–2.70 (m, 2H). HRMS/ESI+ (*m*/*z*): [M + H]^+^ calcd for Pd(C_5_H_8_NO_4_)_2_, 399.0014; found, 399.0044. Anal. Calcd. for Pd(C_5_H_8_NO_4_)_2_: C, 30.13%; H, 4.05%; N, 7.03%. Found: C, 30.29%; H, 4.01%; N, 7.12%.

#### 4.1.19. Synthesis of Bis-(histidinato)palladium(II) (**19**)

Following the standard procedure, 0.0188 g palladium(II) acetate (8.37 × 10^−5^ mol) and 0.0286 g histidine (1.84 × 10^−4^ mol, 2.2 equivalents) were combined. The yield of product was 32.9 mg (0.0793 mmol, 95% yield). Pd(C_6_H_8_N_3_O_2_)_2_ was identified based on the following data: ^1^H NMR (500 MHz, Deuterium Oxide) δ 7.77 (d, *J* = 1.4 Hz, 1H), 7.47 (d, *J* = 1.3 Hz, 1H), 6.99 (s, 1H), 6.97 (d, *J* = 1.3 Hz, 1H), 3.45 (t, *J* = 4.5 Hz, 1H), 3.38 (dd, *J* = 5.6, 3.7 Hz, 1H), 3.22–3.00 (m, 4H). ^13^C NMR (126 MHz, Deuterium Oxide) δ 175.88, 175.73, 136.81, 135.80, 132.77, 132.72, 114.58, 114.28, 52.33, 52.23, 30.41, 30.30. HRMS/ESI+ (*m*/*z*): [M + H]^+^ calcd for Pd(C_6_H_8_N_3_O_2_)_2_, 415.0341; found, 415.0327. Anal. Calcd. for Pd(C_6_H_8_N_3_O_2_)_2_: C, 33.31%; H, 4.19%; N, 19.42%. Found: C, 33.19%; H, 4.25%; N, 19.21%.

#### 4.1.20. Synthesis of Bis-(argininato)palladium(II) (**20**)

Following the standard procedure, 0.0323 g palladium(II) acetate (1.44 × 10^−4^ mol) and 0.0599 g arginine (3.44 × 10^−4^ mol, 2.39 equivalents) were combined. In this case, no clear precipitate was formed. Upon drying overnight on a high-vacuum line 59.3 mg (0.1310 mmol, 91% yield) of a clear yellow varnish was obtained. Pd(C_6_H_13_N_4_O_2_)_2_ was identified based on the following data: ^1^H NMR (400 MHz, Deuterium Oxide) δ 3.30–2.79 (m, 3H), 1.69–1.21 (m, 4H). HRMS/ESI+ (*m*/*z*): [M + 2H]^+^^2^ calcd for Pd(C_6_H_13_N_4_O_2_)_2_, 227.0629; found, 227.0646.

#### 4.1.21. Synthesis of Bis-(lysinato)palladium(II) (**21**)

Following the standard procedure, 0.1393 g palladium(II) acetate (6.20 × 10^−4^ mol) and 0.1996 g lysine (1.37 × 10^−3^ mol, 2.2 equivalents were combined. The yield of product was 228.4 mg (0.5756 mmol, 93% yield). Pd(C_6_H_13_N_4_O_2_)_2_ was identified based on the following data: ^1^H NMR (400 MHz, Deuterium Oxide) δ 3.24 (td, *J* = 7.4, 4.5 Hz, 1H), 2.32 (dtd, *J* = 4.4, 2.2, 1.0 Hz, 2H), 2.20 (d, *J* = 1.0 Hz, 2H), 1.30 (dd, *J* = 6.2, 1.1 Hz, 4H). ^13^C NMR (101 MHz, D_2_O) δ 177.77, 109.99, 66.24, 26.64, 22.23, 22.04. HRMS/ESI+ (*m*/*z*): [M + H]^+^ calcd for Pd(C_6_H_13_N_4_O_2_)_2_, 397.1062; found, 397.1068.

### 4.2. X-ray Crystallography

All single crystals were mounted on a nylon loop on an Oxford diffraction Xcalibur, Sapphire3, Gemini ultra-diffractometer. The crystal was maintained at a low temperature (approximately 100 K) for the duration of the data collection. Data was collected and reduced using CrysalisPro [42] software and then the structure was solved using ShelXS and refined using ShelXL [43]. Graphics were obtained using Olex2 software and Olex2 was also used as an interface to the ShelX suite of software [44]. Complete experimental details for each of the structures can be found in the Appendix A.

## 5. Conclusions

Twenty-one palladium(II) amino acid bis-chelates were synthesized, characterized, and their non-covalent interactions investigated. The amino acids employed came from five general classes based on the chemical nature of their R-groups: aliphatic hydrophobic, aromatic hydrophobic, polar neutral, charged acidic, and charged basic. Each complex coordinates via *N*,*O* chelation to the metal center. Six crystal structures were determined.

This body of work adds to the literature of palladium(II) bis-amino acid complexes. While many of these compounds have been reported previously, it is also the case that they were synthesized at a time when methods of characterization were limited. In this paper, NMR spectroscopy, high resolution mass spectrometry (HRMS), and, for some complexes, single crystal X-ray diffraction greatly expand the information we have on these compounds. While, in the main, there were no real surprises with respect to the overall nature of bis-chelate complexes of amino acids with palladium(II), such as coordination mode and other particulars, there were some new insights into the structures of these compounds. One aspect into which we are continuing to delve is the formation of *cis* vs. *trans* isomers of these compounds with no clear trends emerging. DFT calculations are underway to try to shed some light on this issue.

In specific, the single crystal X-ray diffraction data provide insights into the *inter*molecular hydrogen-bonding interactions exhibited by these compounds. Some conclusions can be made on how bulky amino acid side chain groups can affect these interactions, such as isobutyl groups preventing some direct molecule-to-molecule H-bonding and requiring the intermediacy of water molecules to fulfill H-bonding requirements for packing. However, no real systematics concerning the H-bonding motifs that occur have emerged beyond that and further investigation is warranted. The insights reported here and ones that are produced by future investigations may prove relevant to the studies of the catalytic and bioinorganic properties, especially the biological activity, of this class of palladium(II) complexes. Further studies from our group will probe further into these aspects of palladium amino acid compounds.

## Figures and Tables

**Figure 1 molecules-26-04331-f001:**
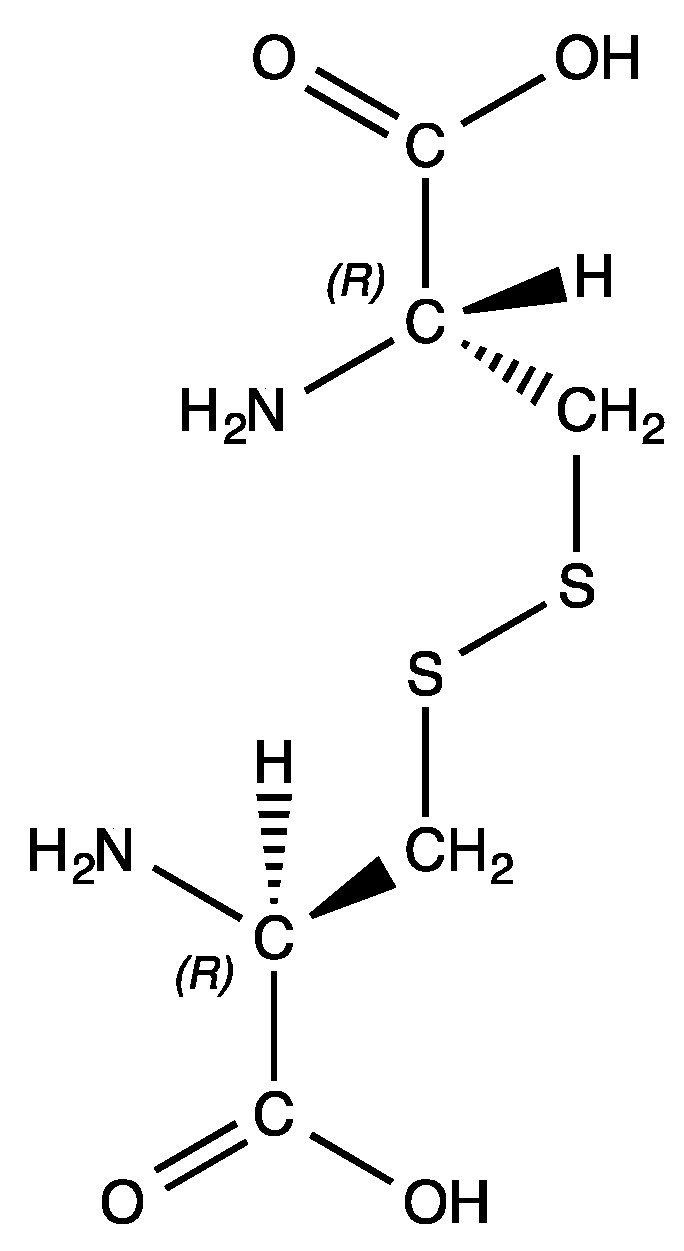
Molecular structure of l-cystine, (R,R)-3,3′-Dithiobis(2-aminopropionic acid).

**Figure 2 molecules-26-04331-f002:**
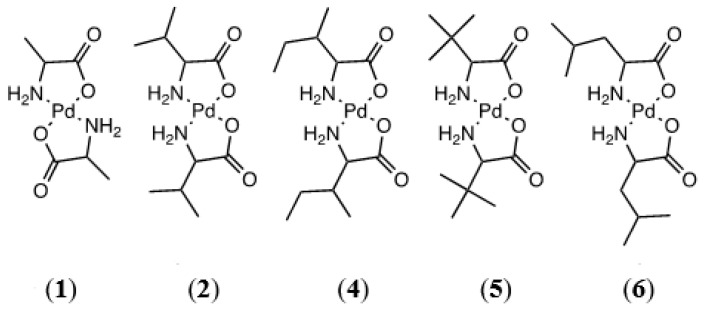
Pd(II) complexes of (*l* to *r*) alanine (**1**), l-valine (**2**), d-valine (**3**, not shown), isoleucine (**4**), *tert*-leucine (**5**), and leucine (**6**).

**Figure 3 molecules-26-04331-f003:**
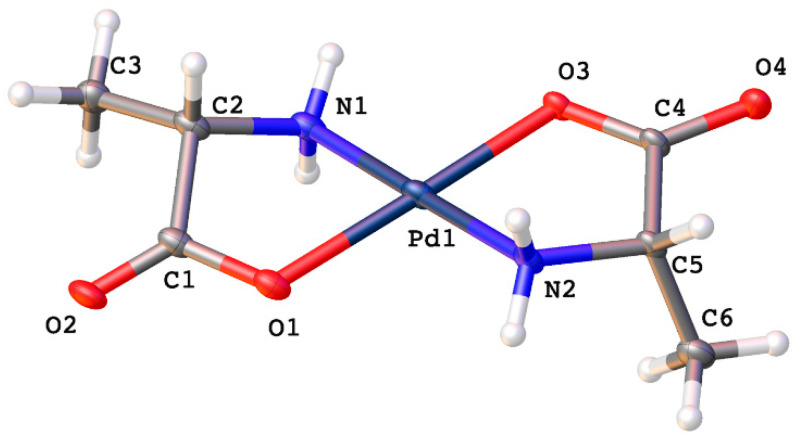
Thermal ellipsoid plot of *trans*-bis-(alaninato)palladium(II) (**1**). Thermal ellipsoids are shown at the 50% probability level. CCDC Ref number: 2055917.

**Figure 4 molecules-26-04331-f004:**
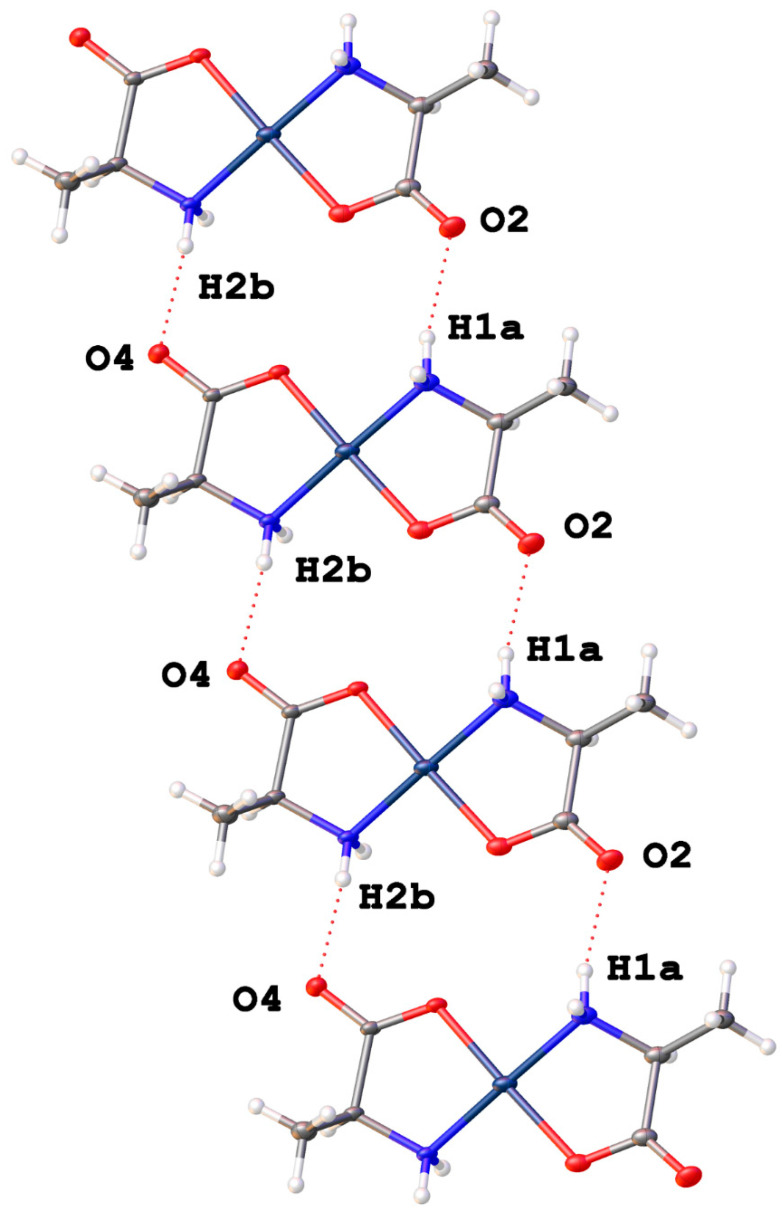
Intermolecular hydrogen bonding interactions observed in the crystal lattice of *trans*-bis-(alaninato)palladium (II) (**1**).

**Figure 5 molecules-26-04331-f005:**
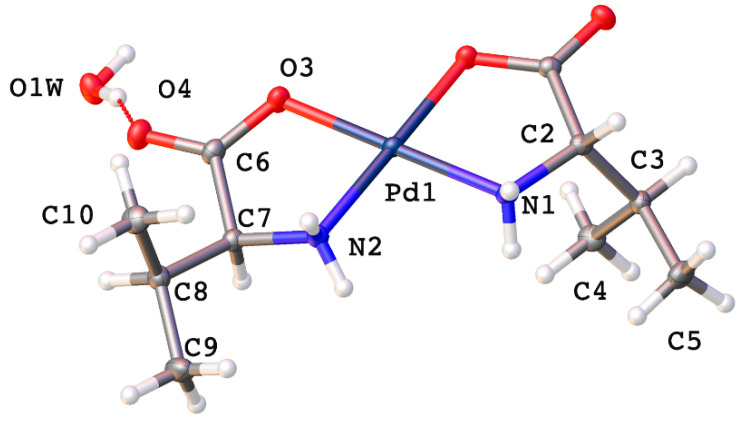
Thermal ellipsoid plot of *cis*-bis-(valinato)palladium(II) (**2**). Thermal ellipsoids are shown at the 50% probability level. CCDC Ref number: 2055914.

**Figure 6 molecules-26-04331-f006:**
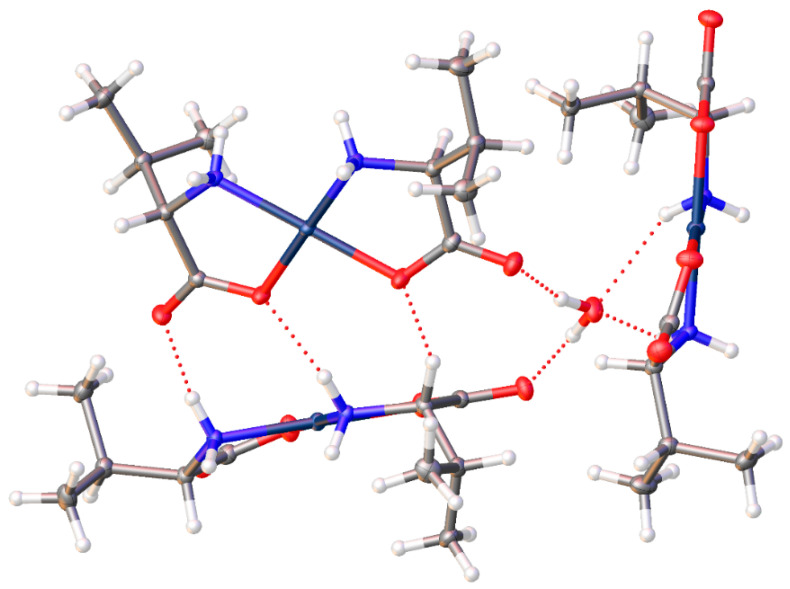
Intermolecular hydrogen bonding interactions observed in the crystal lattice of *cis*-bis-(valinato)palladium (II) (**2**).

**Figure 7 molecules-26-04331-f007:**
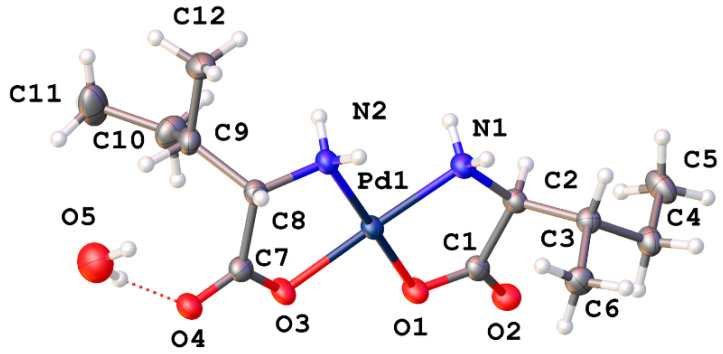
Thermal ellipsoid plot of *cis*-bis-(isoleucinato)palladium(II) (**4**). Thermal ellipsoids are shown at the 50% probability level. CCDC Ref number: 2055915.

**Figure 8 molecules-26-04331-f008:**
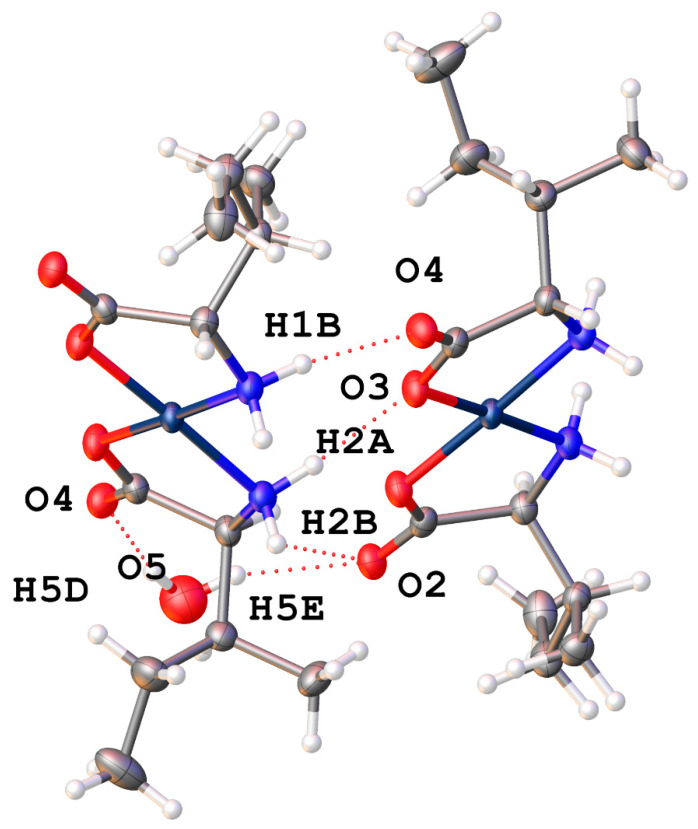
Intermolecular hydrogen bonding interactions observed in the crystal lattice of *cis*-bis-(isoleucinato)palladium (II) (**4**).

**Figure 9 molecules-26-04331-f009:**
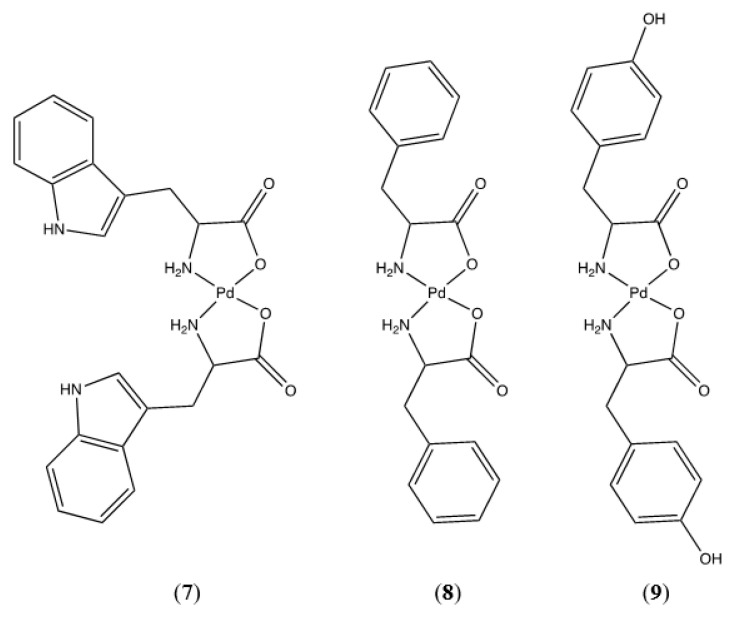
Pd(II) complexes of tryptophan (**7**), phenylalanine (**8**), and tyrosine (**9**).

**Figure 10 molecules-26-04331-f010:**
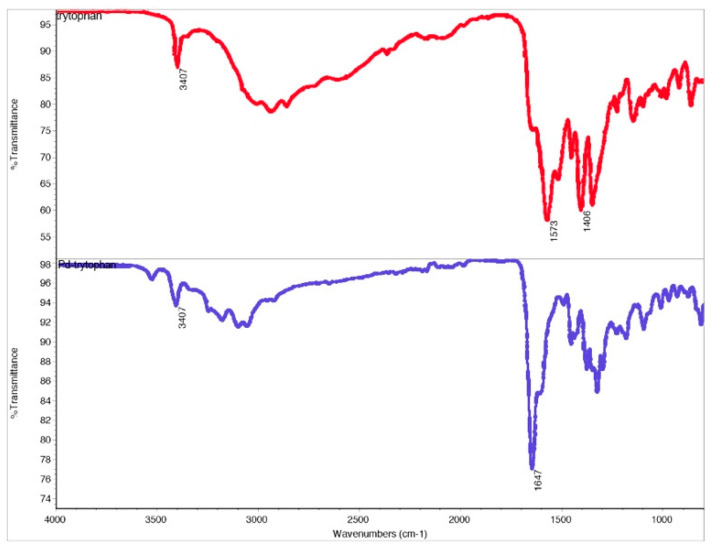
Solid-state FTIR spectral comparison of tryptophan free ligand (**top**, in red) and the bis-(tryptophanato)palladium(II) complex (**7**) (**bottom**, in blue).

**Figure 11 molecules-26-04331-f011:**
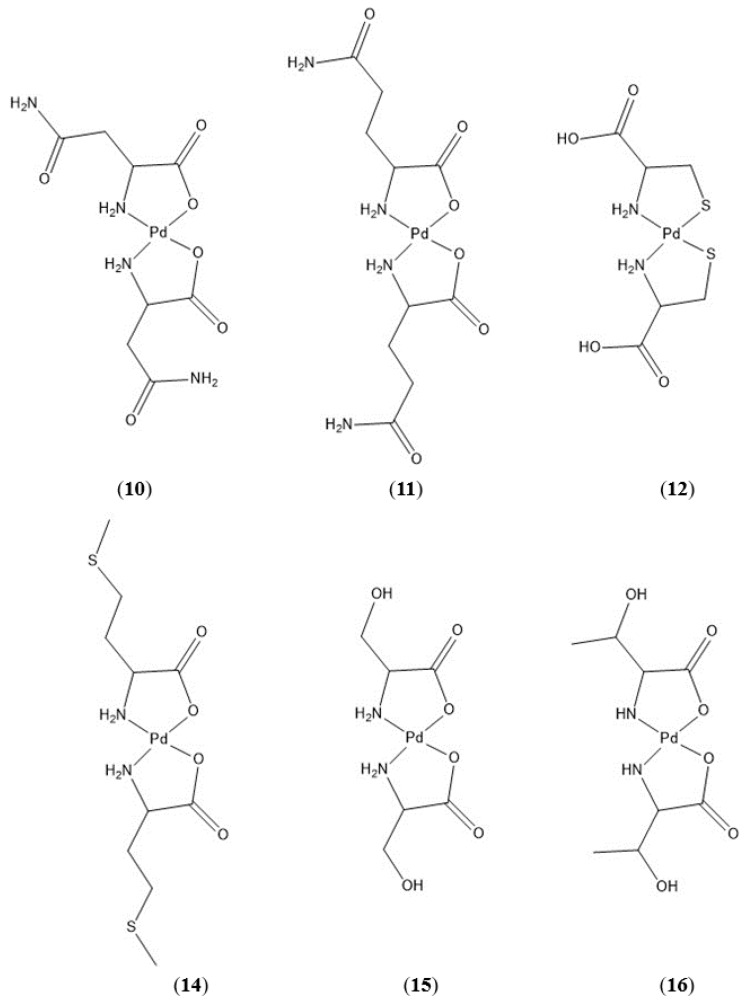
Pd(II) complexes of asparagine (**10**), glutamine (**11**), cysteine (**12**), methionine (**14**), serine (**15**), and threonine (**16**). A complex of cystine (**13**) is not shown because information on such a complex is not available.

**Figure 12 molecules-26-04331-f012:**
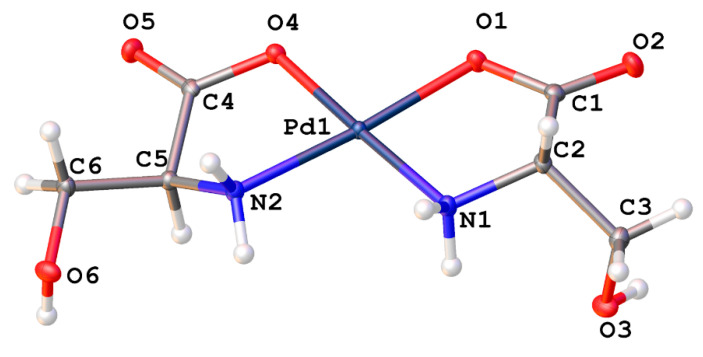
ORTEP plot of *cis*-bis-(serinato)palladium(II) (**15**). Thermal ellipsoids are shown at the 50% probability level. CCDC Ref number: 2055916.

**Figure 13 molecules-26-04331-f013:**
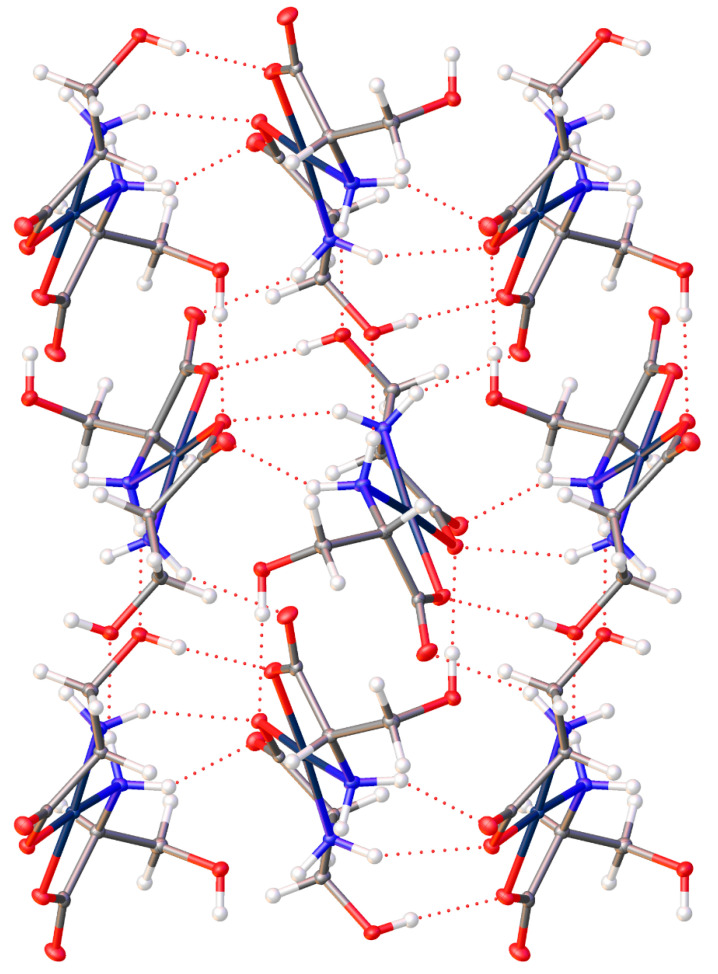
Hydrogen bonding interactions of *cis*-bis-(serinato)palladium(II) (**15**) as viewed down [0 1 0]. Oxygen atoms = red, nitrogen atoms = blue, carbon atoms = grey, palladium atoms = silver. Thermal ellipsoids are shown at the 50% probability level.

**Figure 14 molecules-26-04331-f014:**
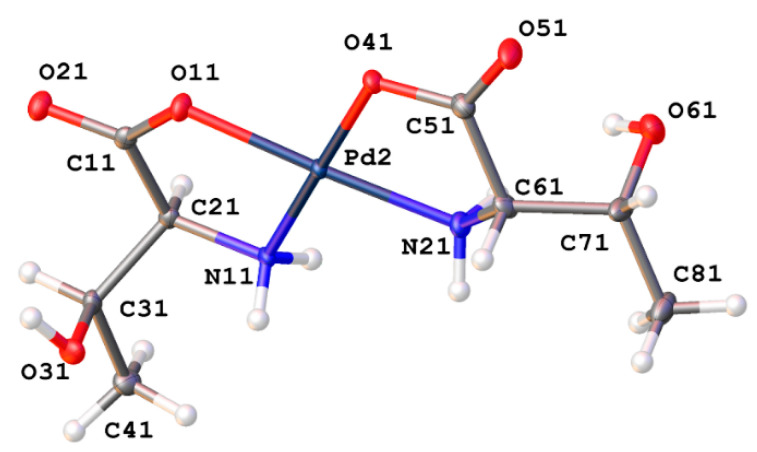
X-ray crystal structure of *cis*-bis-(threoninato)palladium(II) (**16**). Thermal ellipsoids are shown at the 50% probability level. CCDC Ref number: 2055913.

**Figure 15 molecules-26-04331-f015:**
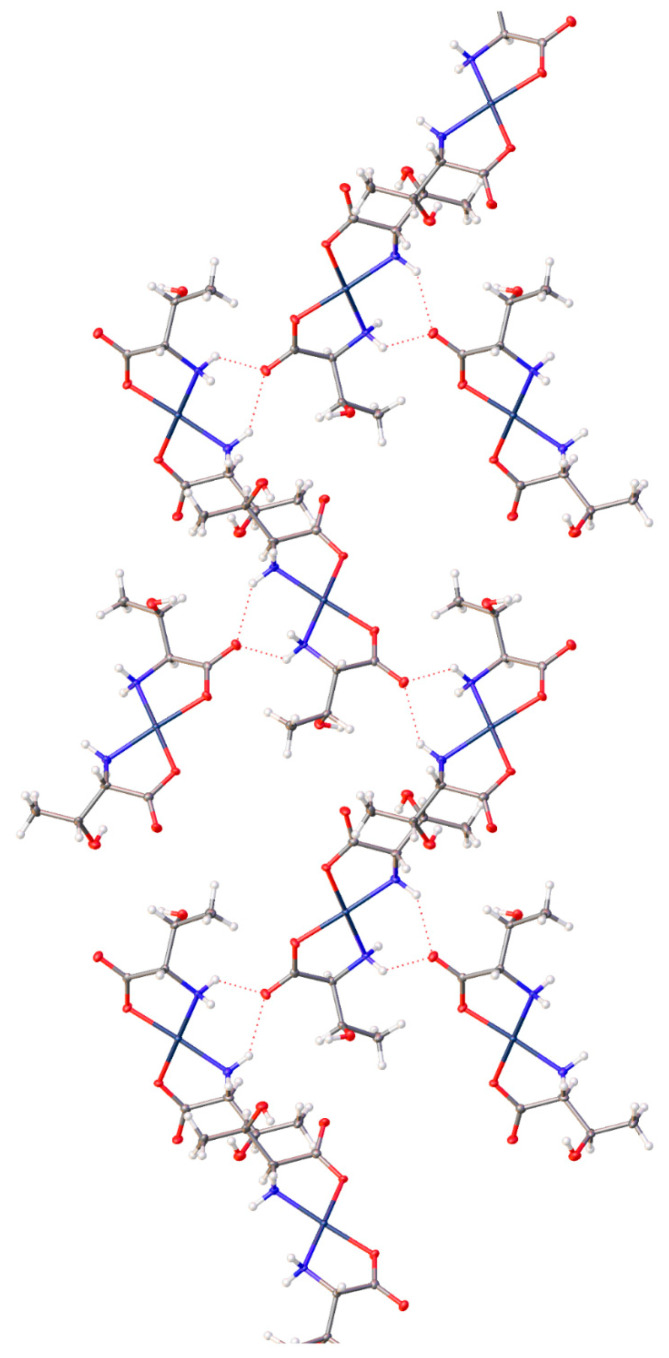
Hydrogen bonding interactions of *cis*-bis-(threoninato)palladium(II) (**16**) as viewed along [0 1 0].

**Figure 16 molecules-26-04331-f016:**
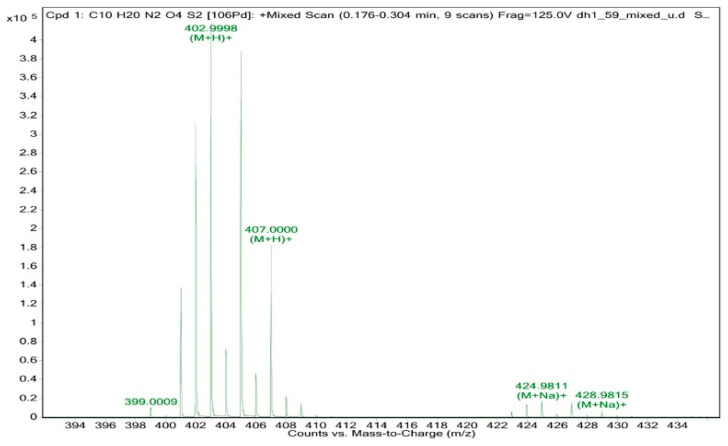
High-resolution TOF mass spectrum of bis-(methioninato)palladium(II) (**14**).

**Figure 17 molecules-26-04331-f017:**
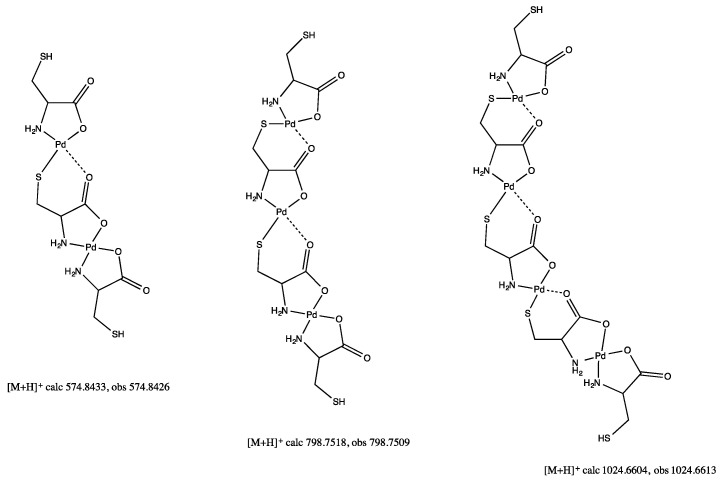
Proposed molecular structures of cysteinato-linked palladium(II) network.

**Figure 18 molecules-26-04331-f018:**
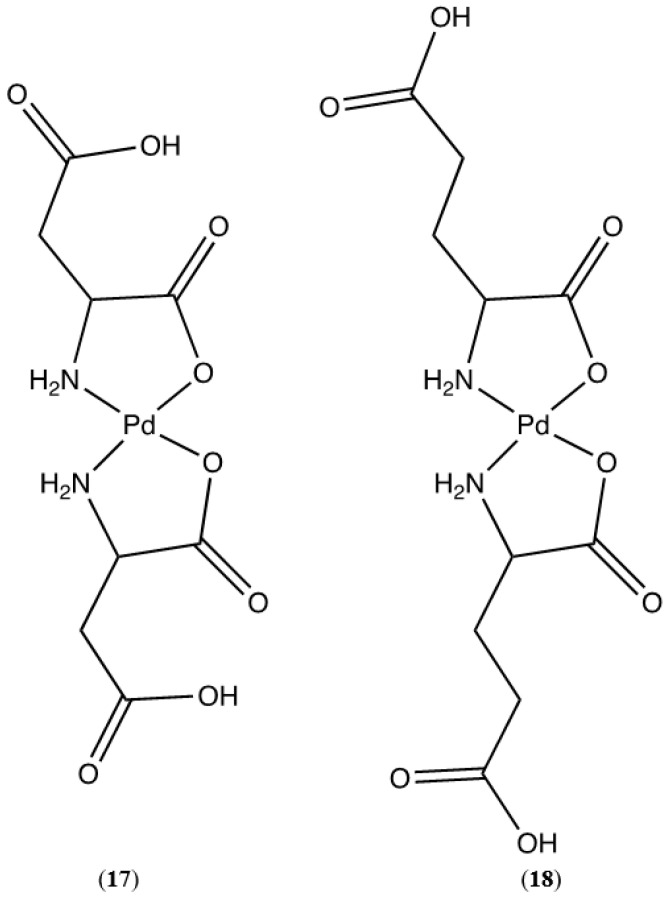
Pd(II) complexes of aspartic acid (**17**) and glutamic acid (**18**).

**Figure 19 molecules-26-04331-f019:**
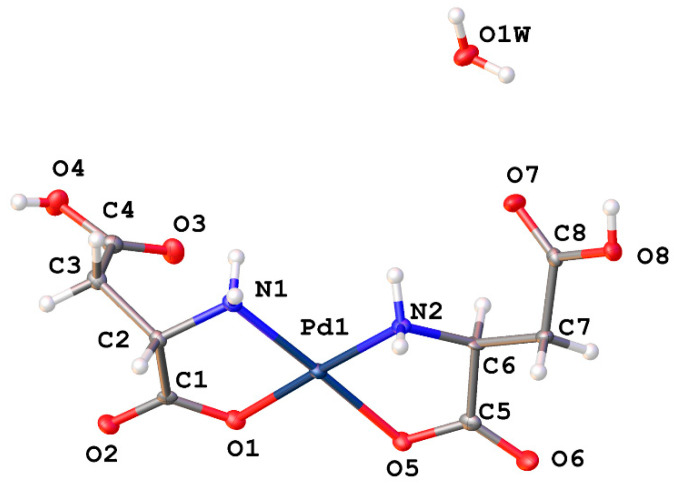
ORTEP plot of the crystal structure of *cis*-bis-((η^2^-*N*,*O*)-aspartate)palladium(II) (**17**). CCDC Ref number: 2055912.

**Figure 20 molecules-26-04331-f020:**
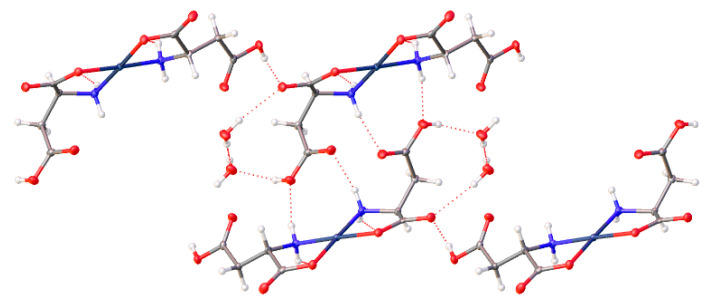
Hydrogen bonding interactions observed in the crystal lattice of *cis*-bis-((η^2^-*N*,*O*)-aspartate)palladium(II) (**17**).

**Figure 21 molecules-26-04331-f021:**
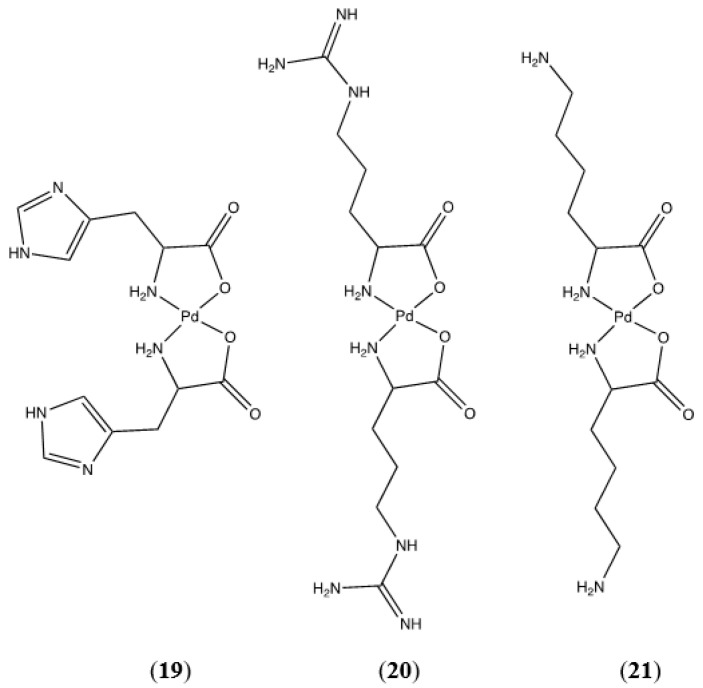
Pd(II) complexes of histidine (**19**), arginine (**20**), and lysine (**21**).

**Figure 22 molecules-26-04331-f022:**
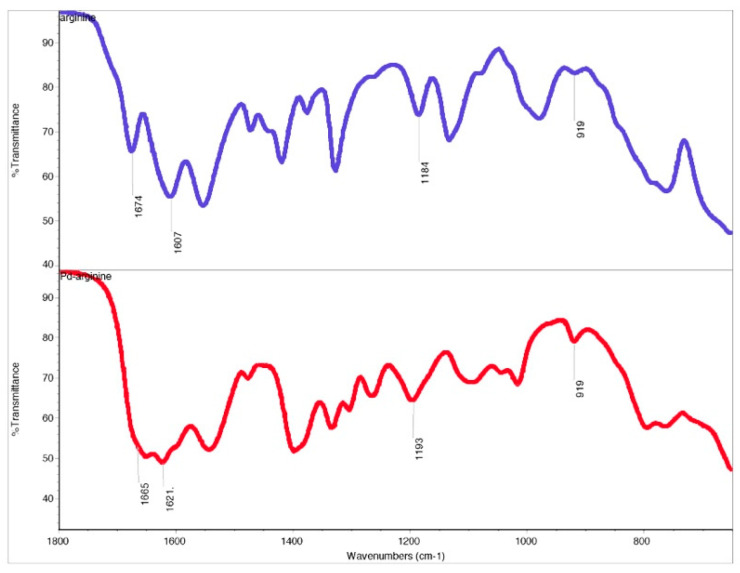
Solid-state FTIR spectral comparison of the free arginine ligand (**top**, in blue) and bis-(argininato)palladium(II) (**20**) (**bottom**, in red) showing the presence of characteristic guanidine C-N stretching vibrations in both molecules.

**Figure 23 molecules-26-04331-f023:**
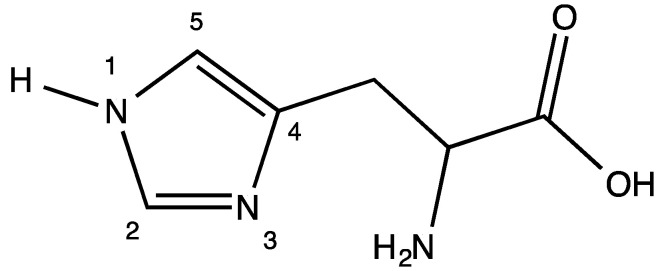
Molecular diagram of histidine. The imidazole group is numbered for clarity.

**Figure 24 molecules-26-04331-f024:**
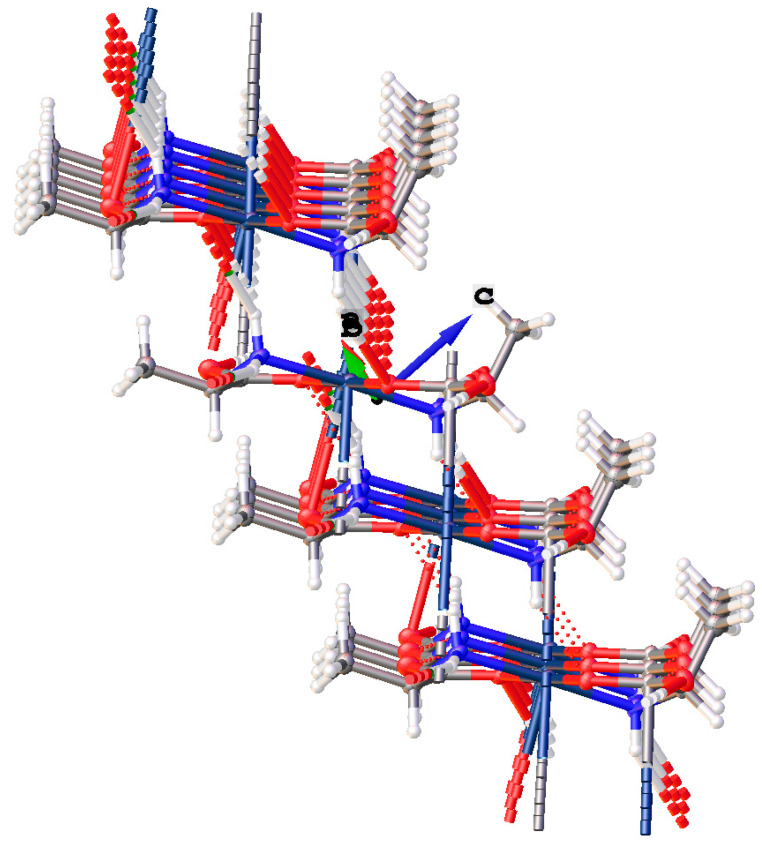
Offset hydrogen-bonded layers in the crystal lattice of complex (**1**).

**Figure 25 molecules-26-04331-f025:**
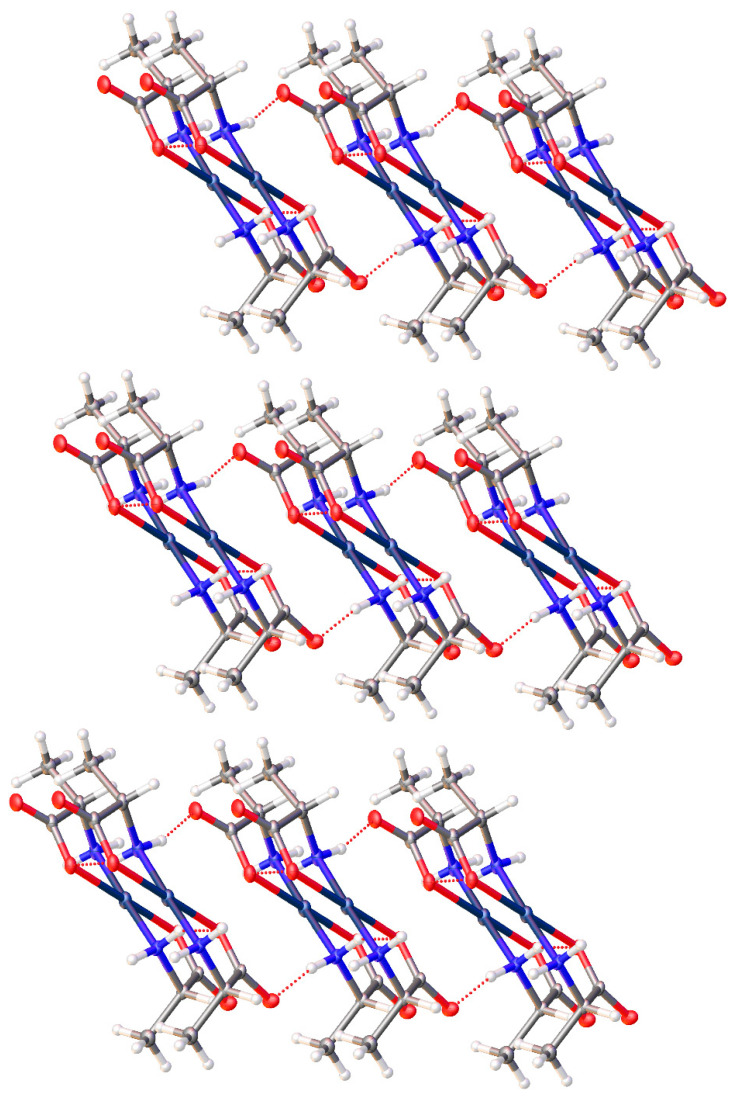
Compound (**1**) lattice view down (1,0,0) showing layers with H-bonding stacked on other layers with van der Waals interactions.

**Figure 26 molecules-26-04331-f026:**
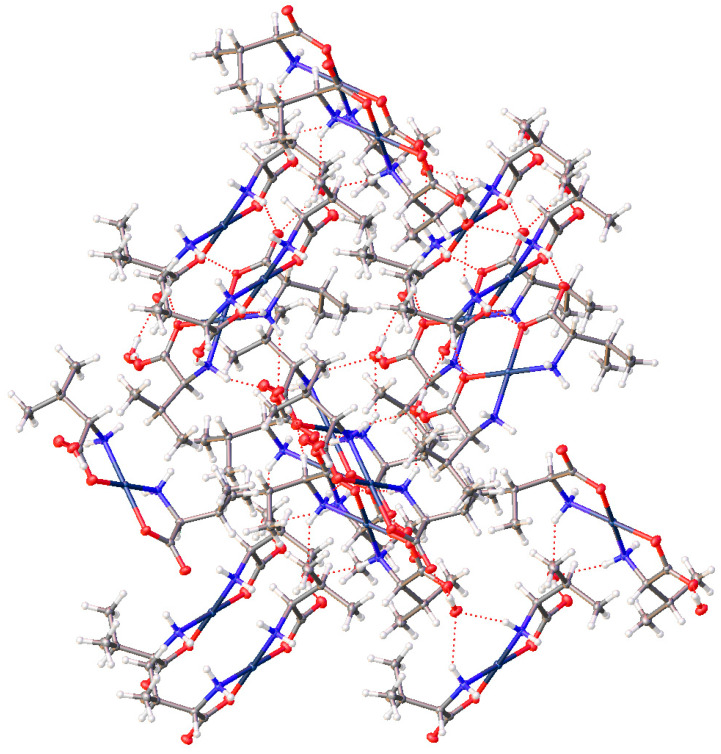
3D H-bonding network in the valine complex, (**2**).

**Figure 27 molecules-26-04331-f027:**
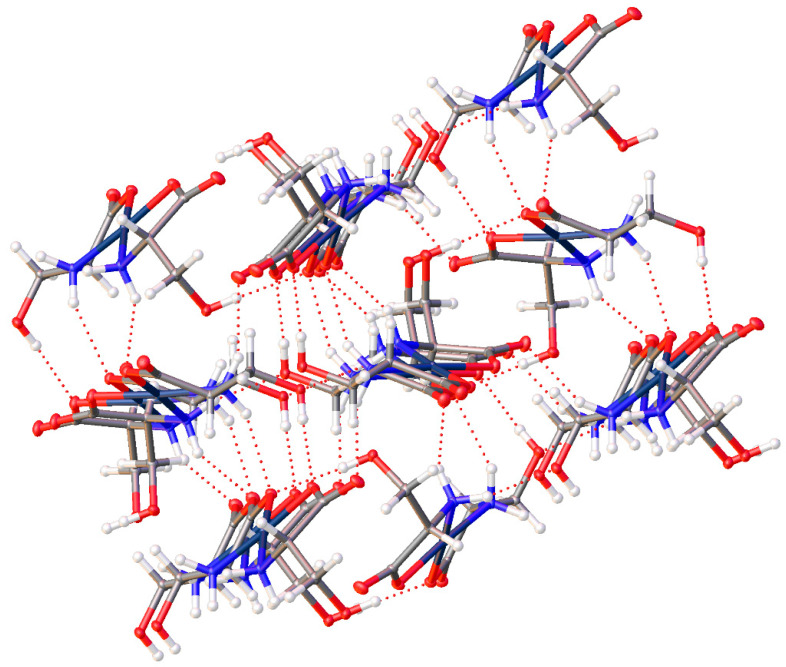
Non-covalent interactions throughout the lattice of the bis-serine complex of Pd(II) (**15**).

**Figure 28 molecules-26-04331-f028:**
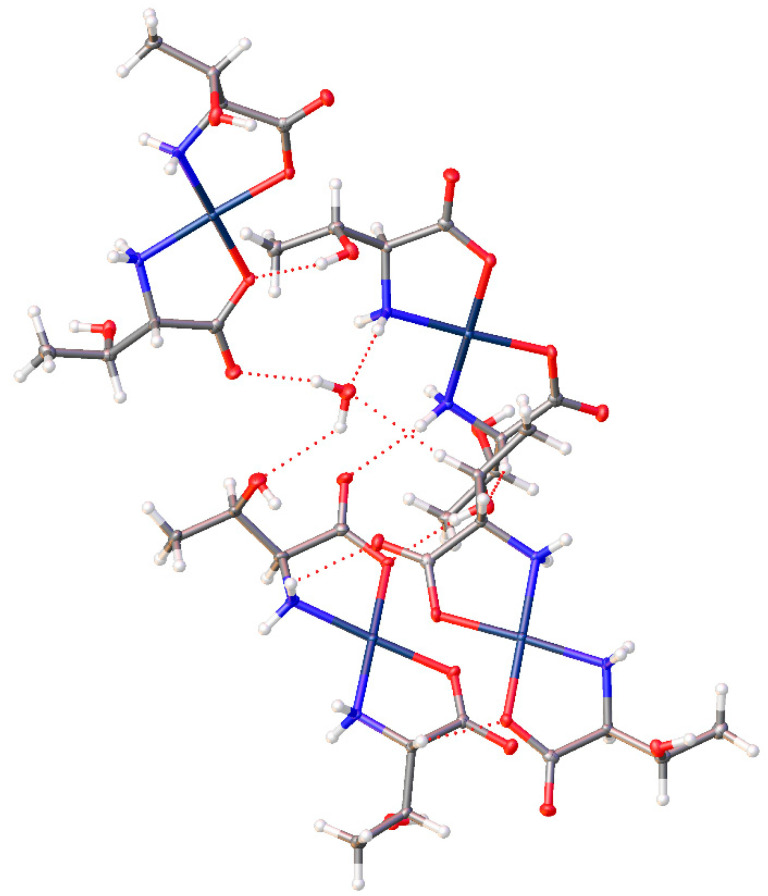
Non-covalent interactions between lattice water and the bis-threonine Pd(II) complex (**16**).

**Figure 29 molecules-26-04331-f029:**
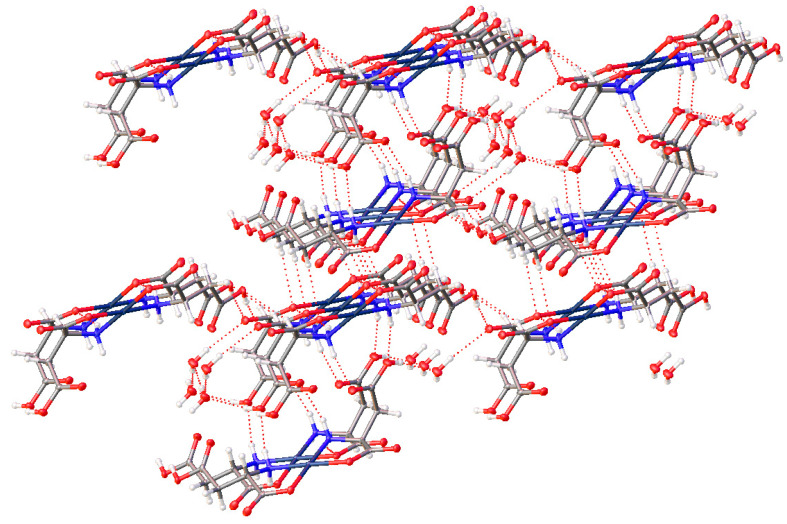
Extended lattice of the aspartic acid complex (**17**) displaying offset layers of Pd centers and water/carboxylic acid layers.

## Data Availability

The data files in cif format for the 6 crystal structures in this paper can be requested from the Cambridge Structural Database. CCDC 2055912–2055917 contains the supplementary crystallographic data for this paper. These data can be obtained free of charge from The Cambridge Crystallographic Data Centre via www.ccdc.cam.ac.uk/structures (accessed on 4 June 2021).

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
