# Peer review of "Synthesis, Characterization, and Non-Covalent Interactions of Palladium(II)-Amino Acid Complexes"

_molecules, 2021, doi:10.3390/molecules26144331_

Round 1

Reviewer 1 Report

Authors reported synthesis and characterization of 21 Pd(II) complexes with amino acids. Among these complexes, the structures of six complexes (1, 2, 4, 15, 16 and 17) have been confirmed by X-ray diffraction analysis, and their intermolecular interactions have been discussed. Nevertheless, the structures of some other complexes were not determined undoubtedly. For instance, complexes with cysteine, cystine and methionine were not isolated to be characterized by spectroscopic NMR and IR techniques, or X-ray diffraction analysis. Considering this, it is very difficult to propose their structures. Also, the proposed structure of complex 12 with cysteine is not correct, because Pd has charge +2, -SH and-COOH are not deprotonated, indicating that the complex should have charge +2, instead of 0, as authors proposed. Moreover, a structure of complex with L-histidine should be confirmed by other methods. It is hard to believe that they did not obtain N,N-chelate (with N3 imidazole coordination). Considering all these facts as well as limited novelty (Pd(II) complexes with amino acids have been reported previously), I think that the manuscript in the present form is not suitable for publication.

Author Response

Authors reported synthesis and characterization of 21 Pd(II) complexes with amino acids. Among these complexes, the structures of six complexes (1241516 and 17) have been confirmed by X-ray diffraction analysis, and their intermolecular interactions have been discussed. Nevertheless, the structures of some other complexes were not determined undoubtedly. For instance, complexes with cysteine, cystine and methionine were not isolated to be characterized by spectroscopic NMR and IR techniques, or X-ray diffraction analysis. Considering this, it is very difficult to propose their structures. Also, the proposed structure of complex 12 with cysteine is not correct, because Pd has charge +2, -SH and-COOH are not deprotonated, indicating that the complex should have charge +2, instead of 0, as authors proposed. Moreover, a structure of complex with L-histidine should be confirmed by other methods. It is hard to believe that they did not obtain N,N-chelate (with N3 imidazole coordination). Considering all these facts as well as limited novelty (Pd(II) complexes with amino acids have been reported previously), I think that the manuscript in the present form is not suitable for publication.

We would like to thank this referee for pointing out a mistake in our structure for figure 12 where the S atoms coordinated to the Pd should not have been protonated.

Beyond that point, we were very clear an open about two aspects of our paper concerning this reviewer's comments.  First, many of these compounds have been mentioned in the literature, but, with no intent to disparage those authors, the techniques available to them at the time did not allow for a complete characterization of the compounds.  In this paper, we have fully characterized quite a number of palladium(II) amino acid compounds including single crystal X-ray structures.  In addition, this paper much more fully discusses the non-covalent intermolecular interactions in the crystal lattices of those compounds, thus fitting into the topic of this special issue of Molecules.

The histidine complex, while not characterized by X-ray diffraction, is characterized by the one method suitable to indicate its mode of coordination - IR spectroscopy.

Finally, we are also very open about the problems with S sidechains and how Pd-S interactions can dominate and cause some intractable materials instead of the discrete complexes.  Asking us to speculate on their structures is not prudent.

Reviewer 2 Report

The authors report the synthesis and characterization of several Pd-aminoacid complexes. The amount of work is impressive and the topic is of potential interest for molecules, however, rather significant improvements are necessary to help the reader with the interpretation of the results.

  1. Conclusions is the weakest place and looks like a combination of abstract with introduction. Please replace the summary of the methods used with actual summary of the results, e.g., what each specific method revealed new, what are the differences in the interactions between different classes, what is the actual expansion of the information reported previously, etc

I must say that Abstract contains more conclusions than Conclusions themselves…

  1. Hydrogen bonding, which is mentioned as one of the most important points of this study (completely agree!) is to a large extent ignored in the discussion. I would expect the authors to compare interactions based on available crystal structures and outline the key factors, differences, dependence on functional groups, strengths, etc.

Figure 20: The authors mention 3 hydrogen-bonded H2O molecules (per f.u., asymmetric unit?) but they are hard to find. Instead, I see hydrogen-bonded water dimers, which are not discussed in the text. Why?

Figure 25 and the text below (bonding in 4, 15, 16) – it is unacceptable to send readers to use external software and investigate bonding picture in the reported compounds! – this is the job of the authors to explain these interactions. If one image is too complex, please split it and use several images, show coordination of the cations and the anions separately, use tables, etc! The authors mention they are disappointed they could not get more SCXRD data but they did not even try to analyze the data they got…

Did the senior researcher approve this text before the submission?

Minor stuff

  1. Please add standard deviations (Esds) for all experimentally determined numbers, particularly bond lengths and angles obtained from SCXRD data.
  2. P12L268, please add ranges and the reference for the literature

Author Response

We thank reviewer 2 for thought-provoking comments about our manuscript.  In the main, we do not fully agree with the reviewer's comments about aspects of how our paper is written, but did find that they prompted us to revise our manuscript in the following ways:

(And, yes, the senior PI on this paper DID review the manuscript and this is he making these responses.)

First, the minor stuff:  esd's have been added and references requested for line 268 (and in other equivalent places) have also been added.

  1. Conclusions is the weakest place and looks like a combination of abstract with introduction. Please replace the summary of the methods used with actual summary of the results, e.g., what each specific method revealed new, what are the differences in the interactions between different classes, what is the actual expansion of the information reported previously, etc

I must say that Abstract contains more conclusions than Conclusions themselves…

-Abstract and conclusions have been rewritten.

  1. Hydrogen bonding, which is mentioned as one of the most important points of this study (completely agree!) is to a large extent ignored in the discussion. I would expect the authors to compare interactions based on available crystal structures and outline the key factors, differences, dependence on functional groups, strengths, etc.

- Much of this section has been rewritten, although not to the level of a major review of the literature.  Discussion on H-bonding has been increased to elaborate on the nature of H-bonding in these complexes and how they change with amino acid functionality.

Figure 20: The authors mention 3 hydrogen-bonded H2O molecules (per f.u., asymmetric unit?) but they are hard to find. Instead, I see hydrogen-bonded water dimers, which are not discussed in the text. Why?

-with all due respect to the reviewer, this comment highlights the reason why we felt that referring the reader to software such as Mercury would be the best way to understand the complexity of these structures.  The number of figures that would need to be included in the manuscript to highlight all aspects of what is happening in this structure (and others) in a 2-dimensional medium would have to be a great number.  Nevertheless, we have added some views to elaborate on the points in the text to make them clearer.

Figure 25 and the text below (bonding in 4, 15, 16) – it is unacceptable to send readers to use external software and investigate bonding picture in the reported compounds! – this is the job of the authors to explain these interactions. If one image is too complex, please split it and use several images, show coordination of the cations and the anions separately, use tables, etc! The authors mention they are disappointed they could not get more SCXRD data but they did not even try to analyze the data they got…

-philosophically, I disagree with the heart of this comment.  I do believe that referring a reader to other software for 3-D analysis is not only acceptable, but prudent, especially in this era when quite powerful software to do it is free as well as easily obtainable.  Nevertheless, we have added some additional figures and additional discussion.

Reviewer 3 Report

The paper will be an excellent contribution to the Molecules journal and definitely should be accepted, just some minor corrections should be done as follows:

line 11, cis and trans words should be put in italics;

lines 30-31: dating back to the 1960s and 1970s - I think it will be good to add some references here in case they are not cited after this sentence;

line 34: time of flight - should be corrected as time-of-flight;

line 43: reported a synthesis characterized - should be mentioned a synthesis of what compound exactly;

line 47: not very clear what is DL?

line 114: x-ray should be corrected as X-ray;

line 193: N,O should be put in italics;

line 214: Section 7 - do the authors mean Supporting Information here?

line 226: The complex of cystine (13) is not shown. - could the authors kindly provide the reason why it's not shown?

line 249: in the literature - I will kindly recommend to give references here if they are not provided elsewhere:

line 251: cis should be put in italics;

line 276: cis should be put in italics;

line 280: cis should be put in italics;

line 284: Sulfur containing should be corrected as Sulfur-containing;

lines 298 and 300: HRMS/esi and HRMS/esi+ - would be good to provide full version of these acronyms, and also esi and esi+ should be capitalized as later in the experimental section;

line 309: N, S - should be corrected as N,S;

line 330: rings respectively - should be corrected as rings, respectively;

line 385: top, in blue - it would be good if the figure might be modified so the blue color of the plot is more pronounced;

lines 419-420: Infantes et al. examined the crystal literature and described various motifs they found[33,34].  - it will be good to mention what motifs;

line 424: hydrogen-bonding should be without hyphen;

line 433: trans should be put in italics.

Author Response

line 11, cis and trans words should be put in italics;  

- this change has been made

lines 30-31: dating back to the 1960s and 1970s - I think it will be good to add some references here in case they are not cited after this sentence;

-Because they are cited in following sentences, we don't believe it is necessary to repeat the citations here.

line 34: time of flight - should be corrected as time-of-flight;

-this correction has been made

line 43: reported a synthesis characterized - should be mentioned a synthesis of what compound exactly;

- the original sentence has been corrected to:  "In 1977 Chernova[6] also reported a synthesis of bis-(alanino)palladium(II) and characterized the product by elemental analysis and molar conductance measurements. " on lines 43 and 44

line 47: not very clear what is DL?

-the sentence in question has been changed to "Bis-(isoleucinato)palladium(II) was prepared by Patel[8] using DL-leucine in 1996 for studying pulse radiolysis and characterized only by elemental microanalysis. " Hopefully the context makes the meaning of DL clearer

line 114: x-ray should be corrected as X-ray;

-this correction has been made

line 193: N,O should be put in italics;

-correction has been made

line 214: Section 7 - do the authors mean Supporting Information here?

-Yes, and "Section 7" has been corrected to "Supporting Information"

line 226: The complex of cystine (13) is not shown. - could the authors kindly provide the reason why it's not shown?  

-a complex of cystine is not shown since evidence for any specific complex is not existent as indicated in the text.  The figure caption has been amended to read "  

"...A complex of cystine (13) is not shown because information on such a complex is not available."

line 249: in the literature - I will kindly recommend to give references here if they are not provided elsewhere:  

-literature references have been added

line 251: cis should be put in italics;

-done

line 276: cis should be put in italics;

-done

line 280: cis should be put in italics;

-done

line 284: Sulfur containing should be corrected as Sulfur-containing;

-done

lines 298 and 300: HRMS/esi and HRMS/esi+ - would be good to provide full version of these acronyms, and also esi and esi+ should be capitalized as later in the experimental section;

line 309: N, S - should be corrected as N,S;

-done

line 330: rings respectively - should be corrected as rings, respectively;

-done

line 385: top, in blue - it would be good if the figure might be modified so the blue color of the plot is more pronounced;

-both figure 10, line 216, and figure 22, line 385 have been modified to make the color differences more pronounced

lines 419-420: Infantes et al. examined the crystal literature and described various motifs they found[33,34].  - it will be good to mention what motifs;

-Infantes has described many different motifs and proposed a nomenclature to describe them, so it is not simple to mention "what motifs".  The following has been added:  "The number of possible hydrogen bonding motifs is quite large and Infantes et al. have proposed a system to describe these complex motifs[33]

line 424: hydrogen-bonding should be without hyphen;

-correction has been made

line 433: trans should be put in italics.

-correction made

Round 2

Reviewer 1 Report

The authors improved the manuscript and answered on the questions raised by the reviewer. 

Reviewer 2 Report

I am glad my thought-provoking comments resulted in the manuscript improvement. The authors probably misinterpreted my comment about Mercury software etc. - the request to add clear images/tables is not to torture the authors but for the majority of readers who will not go for deep analysis but are still interested to have short answers immediately from the paper. We all work with graphical software to analyze the structures so the comment P29L534 is not needed. It is equally (un)needed in any structural paper.

I do not see how conclusions were rewritten. I would expect at least a couple of sentences summarizing the most important point of this study - hydrogen-bonding interactions across the series.  The methods used and speculation about applications of these results better fit to the abstract/introduction and this section shall be limited.

P3L111 Did you really mean Å(ångstrom?) for the chemical shift?

Author Response

I am glad my thought-provoking comments resulted in the manuscript improvement. The authors probably misinterpreted my comment about Mercury software etc. - the request to add clear images/tables is not to torture the authors but for the majority of readers who will not go for deep analysis but are still interested to have short answers immediately from the paper. We all work with graphical software to analyze the structures so the comment P29L534 is not needed. It is equally (un)needed in any structural paper.

-I look forward to having this discussion with this referee over a friendly beverage at any point in the future.  I think we have less of a disagreement in kind rather than substance.  I think I can sum up my point by saying that it is not clear to me that the additions of words and such in the main text will end up convincing a reader who is not already interested to use the 3D tools and so it is better to just direct those readers to those tools without further adieu.   I have deleted the line that this reviewer finds objectionable, but I still feel it may have a place for new entrants into these fields (graduate students, for example) that are not yet aware of some of these tools.  I see questions being asked almost daily in my participation on sites such as "ResearchGate" with students and even some more seasoned practitioners asking about such tools.

I do not see how conclusions were rewritten. I would expect at least a couple of sentences summarizing the most important point of this study - hydrogen-bonding interactions across the series.  The methods used and speculation about applications of these results better fit to the abstract/introduction and this section shall be limited.

-I have rewritten the conclusions.  

P3L111 Did you really mean Å(ångstrom?) for the chemical shift?

-Fixed.  I have no clue as to how that change crept in since that was not the original wording.